# 📏 RULERv2: FROM BASIC RETRIEVAL TO COMPLEX REASONING, A BOTTOM-UP BENCHMARK FOR LONG-CONTEXT EVALUATION

## ABSTRACT

Recent advances in long-context language models have spurred development of diverse benchmarks that often test multiple skills simultaneously, making it difficult to identify specific failure modes. To address this, we introduce RULERV2, a benchmark with systematic difficulty progression from basic synthetic retrieval to complex multi-step reasoning across three domains: multi-key NIAH, multi-value NIAH, and multi-doc QA. We conduct a large-scale evaluation of leading models, including seven closed-source and 26 open-weight models. Our findings reveal a notable performance gap between the two. Critically, we demonstrate that all models, including those claiming million-token context windows, exhibit performance degradation with increasing length, highlighting an unresolved challenge. Our analysis shows that explicit decomposition into a retrieve-then-solve strategy outperforms the implicit, single-step approach, and chain-of-thought reasoning enables models to discover effective decomposition autonomously. Finally, we find that even top-performing open-weight models struggle with fundamental retrieval and copying tasks, leading to degraded performance on more complex problems.[1]

## 1 INTRODUCTION

The rapid advancement of long-context language models has spurred the development of numerous evaluation benchmarks designed to test their capabilities across long sequences of text (Bai et al., 2023; 2024; Yen et al., 2024; Zhang et al., 2024). These benchmarks evaluate a wide range of skills beyond simple literal matching (Kamradt, 2023), including semantic retrieval, summarization, question answering, in-context learning, and coding, with some works developing synthetic tasks to enable evaluation at million-token scales (Vodrahalli et al., 2024; Kuratov et al., 2024). However, these benchmarks suffer from a critical limitation: by testing multiple capabilities simultaneously (retrieval, aggregation, reasoning, etc.), they make it difficult to determine whether failures stem from basic information access or higher-level reasoning, hindering targeted model improvement.

To address this limitation, we introduce RULERV2, a benchmark designed with a bottom-up approach that isolates and tests fundamental long-context capabilities before testing their integration. Unlike existing benchmarks (Li et al., 2025b) showing large degradations from retrieval to reasoning tasks, RULERV2 continuously increases task difficulty between basic retrieval and complex reasoning tasks, enabling step-by-step diagnosis of where and why models fail. RULERV2 comprises of three task domains adapted from RULERV1 (Hsieh et al., 2024), each progressing through four difficulty levels that test a continuous spectrum of abilities: basic (synthetic retrieval), easy (realistic retrieval), medium (retrieve-then-solve), and hard (single-step solve).

1. **Multi-key NIAH:** tests progression from retrieving a single needle among concatenated distractors to solving a single problem among concatenated questions (Liu et al., 2024).
2. **Multi-value NIAH:** tests progression from retrieving multiple needles sharing the same key to counting and copying one of the instances with identical indices (Vodrahalli et al., 2024).
3. **Multi-doc QA:** tests progression from literal document retrieval based on exact content matching to question answering requiring both retrieval and reasoning capabilities (Lee et al., 2025).

---

[1]We release our code at `https://anonymous.4open.science/r/RULERv2`

We conduct a large-scale evaluation of 33 state-of-the-art long-context models, including seven closed-source and 26 open-weight models ranging from 8B to over 100B parameters. Our evaluation reveals critical limitations in current long-context capabilities. All tested models, including those claiming million-token context windows (Meta, 2025; Li et al., 2025a; Yang et al., 2025), exhibit systematic performance degradation as context length increases, challenging current claims of solved long-context understanding. We also find a substantial performance gap between closed-source and open-weight models, with top open-weight models being large, mixture-of-experts (MoE) transformers, while hybrid architectures underperform despite their computational efficiency advantages.

Our detailed analysis yields key insights for improving long-context performance. Explicit task decomposition proves highly effective: retrieve-then-solve strategies consistently outperform direct single-step approaches, while chain-of-thought reasoning enables models to autonomously discover effective decomposition strategies. Test-time scaling methods show mixed results: few-shot demonstrations benefit larger models, but majority voting provides minimal gains despite improved maximum scores across increased number of generations. Most critically, top-performing open-weight models struggle with basic synthetic retrieval tasks when the needle length or quantity increases, revealing a persistent issue of recalling long and scattered information that can propagate and lead to the failure of more complex long-context problems. These findings demonstrate that reliable long-context understanding requires mastering foundational information access before advancing to complex multi-step reasoning over extended contexts.

Our contributions are: (1) We introduce RULERV2, a four-level diagnostic benchmark that systematically increases task difficulty from basic synthetic retrieval to complex multi-step reasoning tasks by layering skills like general knowledge understanding, counting, semantic understanding, or QA reasoning; (2) We provide comprehensive empirical evidence that current models fundamentally still lack robust long-context understanding, spanning from basic information access to more challenging tasks involving million-token contexts; (3) We investigate several approaches to improve performance on complex tasks, including explicit task decomposition through retrieval-then-solve strategies, as well as test-time scaling methods such as few-shot demonstrations, majority voting across parallel generations, and chain-of-thought reasoning.

## 2 THE RULERV2 BENCHMARK

RULERV2 comprises three task domains: multi-key NIAH, multi-value NIAH, and multi-doc QA. Each domain contains four difficulty levels that systematically test long-context capabilities: basic (synthetic retrieval), easy (realistic retrieval), medium (retrieve-then-solve), and hard (single-step solve). This design enables precise diagnosis of model failures. When a model fails at the basic or easy level, the limitation stems from fundamental retrieval abilities. When a model succeeds at medium difficulty but fails at hard, the issue lies in implicit task decomposition rather than underlying skill. All tasks can scale to arbitrary context lengths and support flexible substitution of underlying base datasets. Figure 1 provides an overview of all RULERV2 tasks.

### 2.1 MULTI-KEY NEEDLE-IN-A-HAYSTACK (MULTI-KEY NIAH)

Based on the definition from Hsieh et al. (2024), this task requires models to retrieve a single "needle" (a target piece of information) from a context filled with similar distractors. Prior work has explored variants including phonebook lookups (Jelassi et al., 2024) and JSON key-value retrieval (Liu et al., 2023a). While state-of-the-art long-context models now achieve near-perfect performance on basic retrieval tasks, we use this as a foundation to test progressively more complex capabilities. Our multi-key NIAH extends beyond simple retrieval by requiring models to both locate and solve problems within concatenated question sets.

- **Basic:** Models retrieve numbers associated with query words from key-value pairs (e.g., "magic numbers for AB is: 123"). This isolates pure synthetic retrieval capability.
- **Easy:** Models retrieve complete questions using numerical indices as keys (e.g., "Question 123: What is the capital of France?"). This tests retrieval with realistic, variable-length content.
- **Medium:** Models must first retrieve a question by an index and then it. By explicitly requiring both steps (retrieve-then-solve), the task evaluates whether models can successfully execute guided task decomposition.

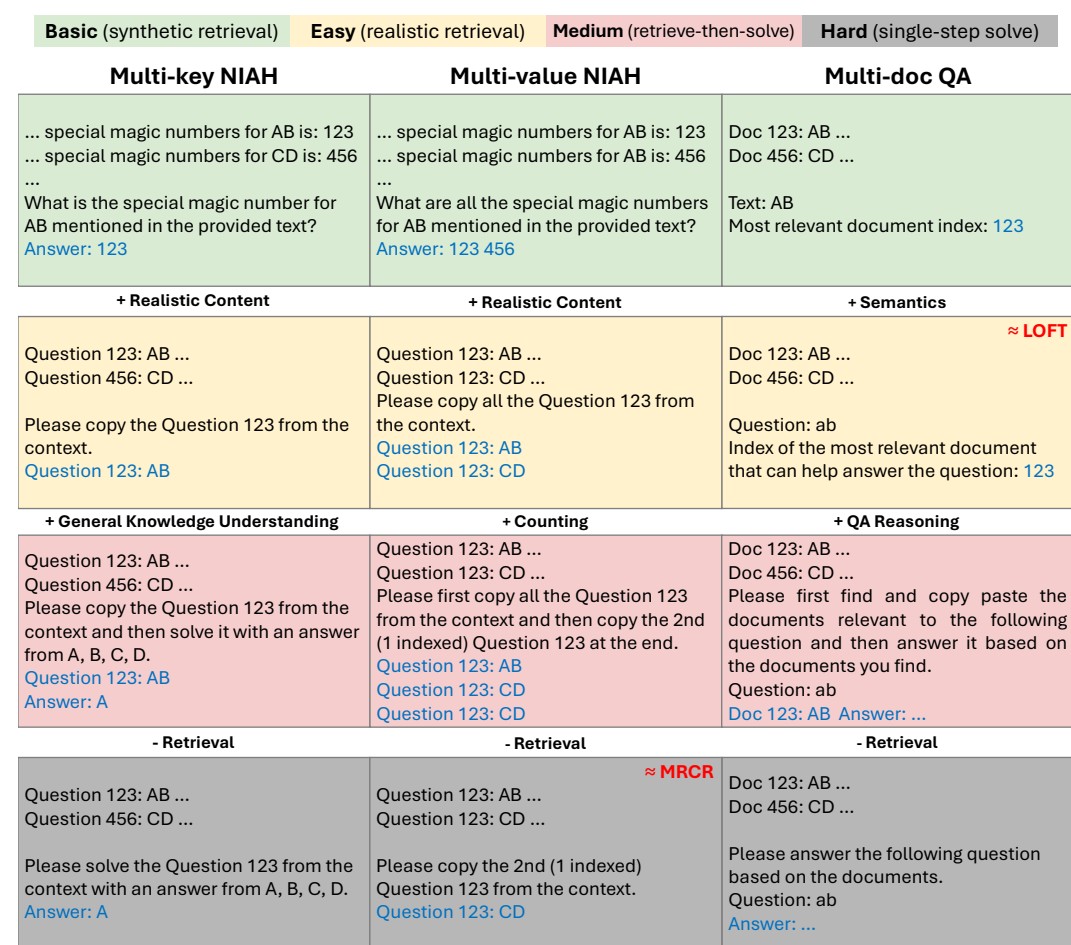

Figure 1: Task examples of RULERV2. We have total 12 tasks among three task domains (multi-key NIAH, multi-value NIAH, multi-doc QA) with four task difficulties (basic, easy, medium, hard). The hard setting of multi-value NIAH is similar to MRCR (Vodrahalli et al., 2024), which requires ordinal position recall, while the easy setting of multi-doc QA is comparable to the text retrieval tasks in LOFT (Lee et al., 2025). See Appendix B for the full task templates.

- **Hard:** Models solve one of the concatenated questions directly without explicit retrieval instructions. Success requires autonomous task decomposition, i.e., recognizing that the complex task requires retrieval followed by reasoning. Performance is upper-bounded by the model's accuracy when the question is presented on its own.

## 2.2 MULTI-VALUE NEEDLE-IN-A-HAYSTACK (MULTI-VALUE NIAH)

Multi-value NIAH extends the basic paradigm by requiring models to find all needles sharing the same key, rather than just one. This tests comprehensive retrieval of non-unique information, a more challenging scenario where multiple relevant items must be located and processed. We progressively add complexity by introducing counting and ordinal selection, creating a controlled version of the Multi-Round Co-reference Resolution (MRCR) task (Vodrahalli et al., 2024). Both tasks require models to identify scattered needles with the same shared keys and use positional information to select the correct answer.

- **Basic:** Models retrieve all numbers associated with a query word from multiple key-value pairs (e.g., if "AB" appears with values 123, 456, and 789, retrieve all three numbers). This tests exhaustive retrieval capability.

- **Easy:** Models retrieve all questions sharing the same numerical index (e.g., all instances of "Question 123: ..."). This introduces realistic, variable-length content to comprehensive retrieval.
- **Medium:** After retrieving all questions with a given index, models must select a specific question by ordinal position (e.g., "the 2nd Question 123"). This combines retrieval with counting and positional reasoning.
- **Hard:** Models directly identify the question at a specified ordinal position without explicit retrieval instructions (e.g., "copy the 2nd Question 123"). This requires autonomous decomposition of the task into: (1) find all relevant questions, (2) order them, and (3) select by position. Our version simplifies MRCR by removing conversational context and formatting requirements while preserving the core joint retrieval-and-counting challenge.

## 2.3 MULTI-DOCUMENT QUESTION ANSWERING (MULTI-DOC QA)

This task tests capabilities central to Retrieval-Augmented Generation (RAG), which appears in various forms in prior benchmarks (Bai et al., 2023; Yen et al., 2024; Lee et al., 2025). RULERV2 progresses from literal, exact-match retrieval to more complex semantic retrieval and question answering.

- **Basic:** Models identify the correct document index when given the exact document content (e.g., given the full text of "Document 5", return "5"). This isolates pure document identification without semantic understanding.
- **Easy:** Models identify documents relevant to answering a question through semantic matching (e.g., given "What is France's capital?" identify documents about Paris or French geography). This tests conceptual relationships between queries and document content.
- **Medium:** Models first retrieve relevant documents by copying them, then answer the question using the retrieved content. This tests guided task decomposition where retrieval and reasoning are separated into distinct steps.
- **Hard:** Models answer questions directly from the full document context without explicit retrieval instructions. Success requires autonomous recognition that the task involves: (1) identifying relevant documents, (2) extracting pertinent information, and (3) creating an answer.

## 3 EXPERIMENTAL SETUP

**Data.** We construct our benchmark using established datasets as base tasks. For multi-key and multi-value NIAH, we use questions from MMLU (Hendrycks et al., 2020) with 5-shot examples. For multi-doc QA, we use HotPotQA (Yang et al., 2018). In multi-value NIAH tasks, we use four needles per target key to test comprehensive retrieval capabilities. We evaluate models across five context lengths: 8k, 16k, 32k, 64k, and 128k tokens. We use 110k tokens to represent 128k score since we need to reserve capacity for model reasoning and output generation. For each context length and task combination, we generate 100 evaluation samples.

**Evaluation Metrics.** We evaluate all tasks using a combined metric that accounts for both exact matches and partially correct responses. For each model response, we compute both recall-based accuracy and word error rate (WER), then take the maximum: $\max(\text{Recall}, 1 - \text{WER})$. This scoring approach captures cases where models provide correct content with minor formatting differences or partial matches, while still rewarding exact matches when they occur.

**Models.** We evaluate seven closed-source and 26 open-weight models, ranging from 7B to 671B parameters across dense transformers, hybrid transformers, and mixture-of-experts (MoE) architectures. We include both general instruction-following models and those with enhanced reasoning capabilities. For inference, we use greedy decoding for standard instruct models and recommended sampling parameters for reasoning models, with 16k token output limits for Chain-of-Thought reasoning. Full model specifications are in Appendix A.

## 4 MAIN RESULTS

**Overall Performance.** Figure 2 shows aggregated RULERV2 performance. Except Grok4, closed-source models outperform open-weight models. Within same model families like Qwen3 (Qwen,

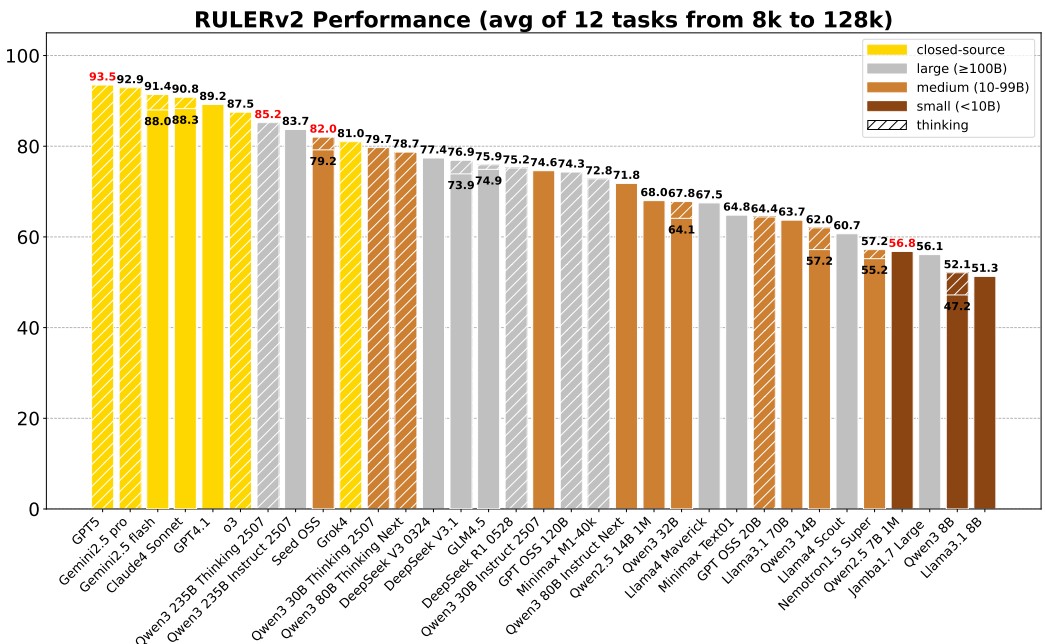

Figure 2: Performance of selected models on RULERV2. Scores are averaged across 12 tasks and five lengths ranging from 8k to 128k. Results with thinking are shown with stripes. The top-performing score for each model class is highlighted in red. Full results are in Table 5 to Table 9.

2025), larger models consistently outperform smaller ones. Top-performing open-weight models are primarily MoE transformers, while hybrid architectures like Llama4 (Meta, 2025), Minimax (Li et al., 2025a), and Jamba (Lenz et al., 2024) underperform expectations, trailing smaller dense transformers. Reasoning models show $3 - 7\%$ improvements when thinking is enabled. Our results indicate that achieving top scores currently relies on a simple scaling approach: increasing model size, extending training length, and incorporating reasoning capabilities.

**Performance Degradation with Context Length.** All models universally struggle as context length increases. As shown in Figure 3 (left), no model maintains stable performance from 8k to 128k tokens. Even top-performing models like GPT5, which claims a 400k context window, show significant degradation. However, open-weight models degrade more severely than their closed-source counterparts. While larger models achieve higher absolute scores, they decline at similar relative rates, suggesting that simply scaling model size does not solve the core length degradation challenge of maintaining performance over longer contexts.

**Performance up to 1M Context Length.** Extending the evaluation to one million tokens reveals further limitations, as seen in Figure 3 (middle). Leading models like Gemini2.5 flash and GPT4.1 experience a performance drop of approximately 15% at 1M tokens. Open-weight models face greater challenges; for example, Qwen3 235B 2507 falls sharply beyond 256k tokens, indicating that current length extrapolation techniques like dual chunk attention (An et al., 2024) and attention temperature scaling (Peng et al., 2023) are insufficient for ultra-long contexts. Hybrid architectures also show mixed results: Llama4 and Qwen3 80B Next degrades sharply, while Minimax is stable but has low overall scores. This suggests that computationally efficient hybrid models have not yet matched the long-context performance of full-attention transformers.

**Comparison between RULERV1 and RULERV2.** A comparison with RULERV1 highlights the progress in long-context capabilities. As Figure 3 (right) shows, current models now achieve saturated scores and pass a established threshold on RULERV1, indicating it less effective for differentiating modern models. On the other hand, RULERV2, addresses this by providing more room for improvement. On RULERV2, performance systematically decreases as context length and

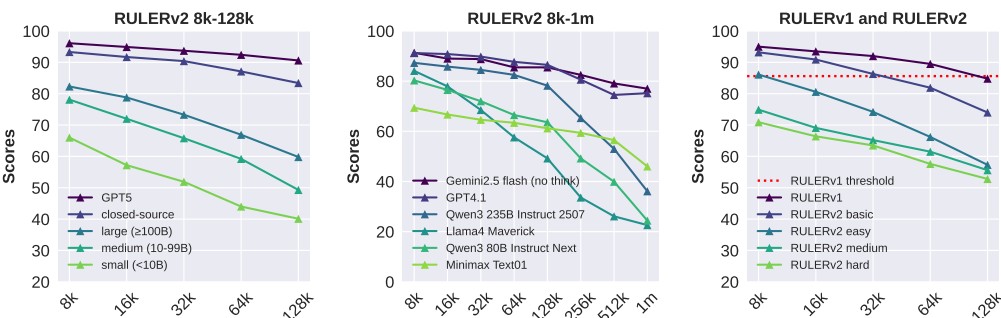

Figure 3: (**Left**): Comparison of different model sizes from lengths 8k to 128k. (**Middle**): Comparison of models claiming 1m context length. (**Right**): Comparison of RULERV1 and RULERV2.

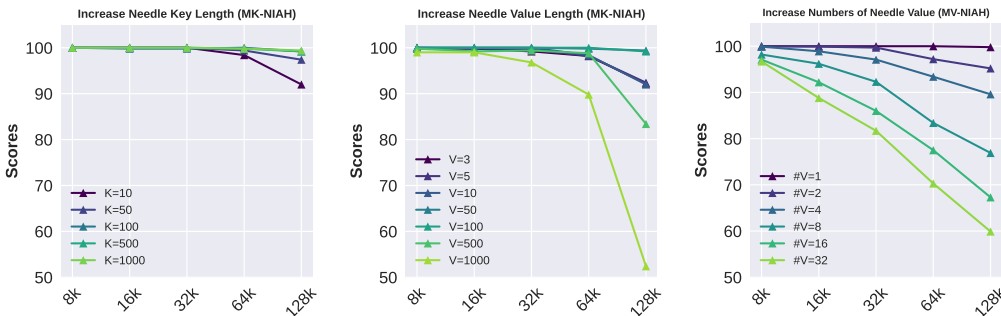

Figure 4: Needle-in-a-haystack variants by increasing needle key length (**left**) and needle value length (**middle**) in multi-key NIAH as well as the numbers of needle value (**right**) in multi-value NIAH. We use model `Qwen3 235B Instruct 2507` for this analysis. Additional results can be found in Appendix C.1.

task difficulty increase. By progressing from basic retrieval to complex reasoning, it challenges even top-performing models where RULERV1 show saturation. This confirms RULERV2's utility in measuring fundamental long-context skills and identifying clear areas for improvement.

## 5 ANALYSIS

### 5.1 NEEDLE-IN-A-HAYSTACK VARIANTS

Following RULERV1, which proposed several NIAH variants by altering the type and quantity of needles and haystacks, we analyze the impact of varying the needle's key length, value length, and the number of values associated with a single key. The realistic retrieval (easy levels) of our multi-key and multi-value NIAH tasks can be viewed as a practical form of increasing the needle's value length.

**Increase Needle Key Length.** To analyze the effect of key length, we use numbers with an increasing number of digits as the needle key, with results shown in Figure 4 (left). We observe performance degradation when using keys with 10 and 50 digits, but the model performance is stable for larger key lengths. This suggests that longer needle keys are actually easier for the model to locate, likely because they provide more distinctive patterns that reduce ambiguity and false matches with other subsequences in the context.

**Increase Needle Value Length.** As shown in Figure 4 (middle), increasing the needle value length to 500 or 1000 digits leads to significant performance degradation. This occurs because the model struggles to accurately copy extremely long and contiguous sequences from the context. Interestingly, we also observe a performance drop for very short values (3, 5, and 10 digits). This may be because

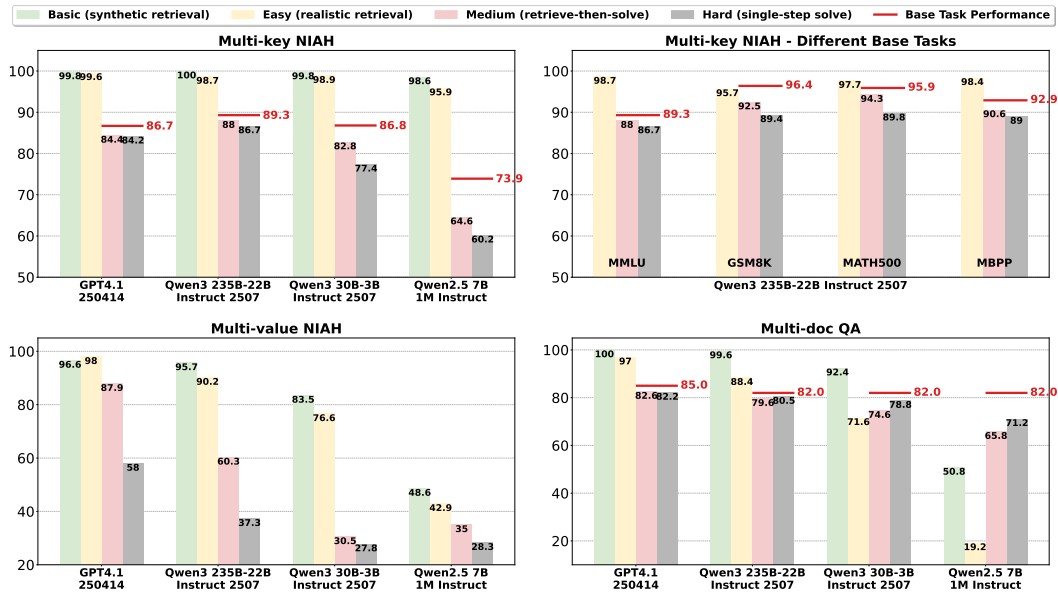

Figure 5: We analyze four task difficulties in three task domains, using four instruct models from each model sizes. Base task performance is the score without adding any distractors.

under a fixed context length budget, shorter needle values result in higher needle density, leading to substantially more distractors within the context. This increased distractor density makes the retrieval task more challenging despite the individual needles being shorter.

**Increase Numbers of Needle Value.** Our default Multi-value NIAH task uses four needles, but other work has explored different configurations (Comanici et al., 2025; OpenAI, 2025a). In Figure 4 (right), we show that increasing the number of needles leads to progressive performance degradation, as it requires the model to attend to multiple scattered locations within the context. This analysis indicates that current models cannot reliably retrieve all relevant information, making this a key ongoing challenge. We hypothesize that the root cause may stem from fundamental limitations of the attention mechanism, a direction that requires further investigation.

## 5.2 Task Difficulties from Basic to Hard

RULERV2 comprises 12 tasks across three domains and four difficulty levels. We analyze the performance of four different model sizes across these difficulties, plotting the results in Figure 5.

**Multi-key NIAH.** In Figure 5 (top left), all models achieve near-perfect scores on the basic and easy levels, though 7B model exhibits a clear degradation from basic to easy. The downward trend in scores with increasing difficulty suggests that failures may propagate to more complex problems. At the medium and hard levels, all models show a performance drop relative to base task performance (i.e., their short-context MMLU score). We observe a similar trend when substituting other base tasks like GSM8K (Cobbe et al., 2021), MATH500 (Hendrycks et al., 2021), and MBPP (Austin et al., 2021), with the degradation being particularly severe for math-related benchmarks (top right). Notably, performance on the medium level (retrieve-then-solve) is consistently higher than on the hard level (single-step solve). This suggests that this task can be effectively decomposed, and good retrieval is a prerequisite for reliably solving a specific question embedded within a larger context.

**Multi-value NIAH.** As shown in Figure 5 (bottom left), open-source models perform poorly under the basic level, with the smaller 7B model achieving only a 48.6% accuracy. Even GPT4.1, which demonstrates near-perfect retrieval, struggles to jointly perform retrieval and counting in the hard setting, achieving only a 58% accuracy. This finding helps explain why MRCR (Vodrahalli et al.,

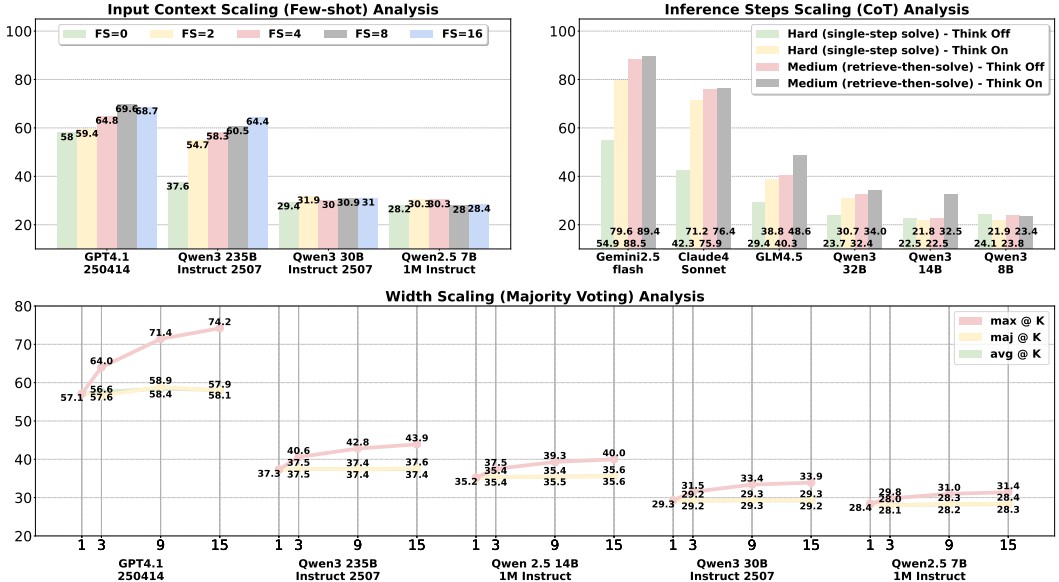

Figure 6: Results of test-time compute scaling including input context scaling (**top left**), inference steps scaling (**top right**), and width scaling (**bottom**). All the scores are averaged across lengths 8k to 128k and evaluated on the hard level of multi-value NIAH task.

2024) is challenging: failures can stem from the retrieval (basic and easy), the counting (medium), to the implicit task decomposition (hard). Decomposing the task into explicit retrieval and then counting under medium setting results in a substantial performance improvement for all models compared to the harder setting. Analysis of model outputs reveals that successful medium-level responses typically first enumerate all relevant questions before performing the ordinal selection, while hard-level failures often result from models attempting to count implicitly without explicit enumeration.

**Multi-doc QA.** The results in Figure 5 (bottom right) show that open-weight models cannot perfectly solve even the basic synthetic retrieval task. When semantic understanding is required at the easy level, all models' performance degrade, likely due to failures in understanding latent associations between query and document beyond literal keyword matching (Modarressi et al., 2025). At higher difficulty levels, all models fail to match the baseline task performance. Counterintuitively, for smaller models, hard-level scores sometimes exceed medium-level scores. Manual inspection of outputs suggests that smaller models, even without explicit retrieval instructions, often answer with directly extracted spans. In contrast, larger models tend to paraphrase the retrieved information, leading to more mismatches under the exact-match (Rajpurkar et al., 2016) metric.

## 5.3 SCALING TEST-TIME COMPUTE

We explore three test-time compute scaling methods: (1) scaling input context via few-shot demonstrations, (2) scaling inference steps via Chain-of-Thought reasoning, and (3) scaling width via majority voting. We test these methods under the most challenging task, multi-value NIAH (hard).

**Scaling Input Context.** We scale the number of few-shot demonstrations from 0 to 16. As shown in Figure 6 (top left), performance improves for larger models, while smaller models show no improvement over their zero-shot score. This suggests a capacity threshold: models require sufficient parameters to effectively learn complex retrieval and counting patterns from in-context examples.

**Scaling Inference Steps.** We evaluate models with reasoning capability across both medium and hard difficulty levels. Figure 6 (top right) confirms the benefit of generating a chain of thought before the final answer, with all except the smallest models showing improvements. Crucially, performance on the hard task with thinking approaches the performance on the medium task without thinking,

suggesting that explicit reasoning allows the model to autonomously decompose complex tasks. Manual analysis of generated reasoning chains reveals that models consistently follow a retrieve-then-count pattern: first enumerating all relevant information, and then performing ordinal selection. This mirrors our benchmark decomposition (medium setting), demonstrating that reasoning-capable models can autonomously discover problem-solving strategies.

**Scaling Width.** We generate 1 to 15 parallel responses and analyze maximum, majority, and average scores in Figure 6 (bottom). Majority voting provides little improvements, as models consistently produce similar incorrect responses rather than diverse attempts. However, maximum scores across all generations steadily increases, especially for larger models. For instance, the performance gain from 1 to 15 generations is 6.3% for the largest open-weight model versus 3.0% for the smallest. This suggests that while the primary prediction is stable, larger model possess a broader solution spaces and occasionally generate correct responses which could be identified through more selective sampling strategies.

## 6  RELATED WORK

Existing long-context benchmarks typically pursue comprehensive evaluation by testing diverse capabilities simultaneously, including retrieval, question answering, summarization, in-context learning, and coding. Early benchmarks were limited to shorter contexts (Shaham et al., 2023; An et al., 2023; Bai et al., 2023; Dong et al., 2023; Li et al., 2023), but recent work has extended evaluation to contexts ranging from 128k to over a million tokens (Zhang et al., 2024; Hsieh et al., 2024; Yen et al., 2024; Bai et al., 2024; Lee et al., 2025). Beyond these multi-task benchmarks, researchers have developed synthetic tests to isolate specific long-context capabilities. The widely used "Needle-in-a-Haystack" (NIAH) test (Kamradt, 2023) evaluates fact retrieval from extremely long documents. Other synthetic benchmarks include MRCR (Vodrahalli et al., 2024) for ordinal position recall, NoLiMa (Modarressi et al., 2025) for latent association inference, BABILong (Kuratov et al., 2024) and GSM-$\infty$ (Zhou et al., 2025), for multi-step reasoning, Sequential-NIAH (Yu et al., 2025) for sequential information extraction, NeedleThreading (Roberts et al., 2024), for following threads of information, CountingStars (Song et al., 2025), for multi-evidence counting, NeedleBench (Li et al., 2025b), for progressively testing information-sparse retrieval to information-dense ancestral tracing. These tasks share a common foundation that they introduce novel long-context reasoning tests by fundamentally building on top of retrieval capabilities (Goldman et al., 2024). These works have demonstrated that even though model can solve retrieval tasks, the model has limitations to solve hard reasoning with retrieval tasks. However, by testing combined skills, existing benchmarks obscure whether failures stem from basic information access, higher-level reasoning processes, or implicit skill decomposition. This limitation hinders targeted model improvement. We address this gap by developing RULERV2 with a systematic bottom-up approach that isolates retrieval as a foundational skill and continuously layers additional capabilities. Compared to prior works mainly focus on finding degradations by building reasoning tasks from solved retrieval tasks (NIAH), our design enables precise diagnosis of model limitations to clear understand how a small task change can lead to degradations in hard tasks, highlighting areas requiring focused improvement.

## 7  CONCLUSION

We introduced RULERV2, a systematic bottom-up benchmark that progressively increases task difficulty from basic synthetic retrieval to complex multi-step reasoning across three key domains. Through comprehensive evaluation of 33 long-context models, we uncovered several critical limitations in current long-context capabilities that challenge existing claims of solved long-context understanding. Our evaluation reveals that all models, including those claiming million-token context windows, exhibit performance degradations as task difficulty and context length increase. Most importantly, our analysis demonstrates that even top-performing open-weight models still struggle with fundamental retrieval and copying tasks, which are skills that serve as prerequisites for more complex reasoning. This finding suggests that the path to reliable long-context AI requires a foundation-first approach, where mastering basic information access capabilities precedes attempts at complex multi-step reasoning. Our analysis shows that explicit task decomposition through retrieve-then-solve strategies consistently outperforms single-step approaches, demonstrating the value of breaking

complex tasks into manageable components. Additionally, our exploration of test-time compute scaling shows that chain-of-thought reasoning enables models to autonomously discover effective problem-solving strategies, with performance approaching that of explicitly decomposed tasks for several models. RULERV2 addresses a critical gap in long-context evaluation by systematically isolating fundamental capabilities, providing a rigorous framework for diagnosing model limitations and measuring progress on the core skills that underpin reliable long-context understanding.

## 8 LIMITATIONS

This study has several limitations that we have considered and describe in details below.

**Lack of correlation with realistic long-context tasks.** All of our tasks are designed to evaluate long-context skills synthetically. This is because there are no realistic long-context tasks that are easy to scale to millions of tokens and can be automatically evaluated without manual checking. While we emphasize our benchmark as a convenient check to verify long-context capabilities, we still need to test models in realistic settings that are closer to how they would be truly used. These settings would require multiple capabilities beyond retrieval, such as question answering from books (Karpinska et al., 2024; Fiction.liveBench, 2025), processing code repository (Liu et al., 2023b; Bogomolov et al., 2024), and many-shot in-context learning (Bertsch et al., 2024; Zou et al., 2024). While naturalistic benchmarks (Bai et al., 2024) are necessary, we position RULERV2 as a diagnostic complement to analyze the long-context model limitations.

**Lack of a clear definition of fundamental skills.** Our bottom-up benchmark is built to progressively increase in difficulty, starting with solved retrieval task and moving on to the reasoning ability. However, we need a clearer definition of the fundamental skills required for other long-context tasks and decide whether they relate to retrieval. For example, we view aggregation as an ability building upon retrieval (Liu et al., 2025), our definition is to process pieces of information across multiple locations, recognize the connections between these pieces, and synthesize into coherent higher-level representations. Actually, our multi-value NIAH task can be regarded as an aggregation task, but we only emphasize retrieval ability in this work. Therefore, more studies are needed to break down difficult and complex long-context tasks into fundamental skills. This would help us better analyze failures and understand the limitations of current long-context language models.

**Potential data contamination.** Since we have used MMLU (Hendrycks et al., 2020) and HotPotQA (Yang et al., 2018) as the base task, we may have data contamination issue that models have already memorized the questions or documents. This concern can be significant only on the multi-key NIAH easy task because we need to copy-paste the questions from the context. For other tasks, memorization can hardly solve them because we test retrieving multiple instances in multi-value NIAH and document index searching in multi-doc QA. To mitigate the high score saturation in our benchmark, we also include base task performance in Figure 5. This score measures the best-case performance to solve the MMLU or HotPotQA task in a zero-distractor setting, which includes any benefits from memorization. The significant gaps between base task performance and the scores on medium or hard tasks show a clear failure of retrieval and task decomposition. The model may know the answer to the memorized question, but it fails to find the correct question or document.

**Hollowed-Out Reasoning.** We intentionally chose the saturated short-context tasks like MMLU as our base task instead of using some hard synthetic reasoning tasks like prior works (Li et al., 2025b; Kuratov et al., 2024). It is because we want to verify the propagation of retrieval issue from basic level to hard level. We need a easy task to differentiate confounding errors. On the other hand, selecting a very difficult reasoning task as the base task may cause the score to be pretty low and hard to analyze the fundamental skill failures. However, this decision will reduce the claim of complex reasoning since it is no longer test reasoning with retrieval but only autonomous task decomposition with retrieval. Therefore, we remain the flexibility to change base task once we see the saturation in our benchmark.

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

# A    MODELS

In this work, we select in total 33 models for evaluation including seven closed-source and 26 open-weight models (listed in Table 1). For open-weight models, we have 12 large size ($\geq$100B), 11 medium size (10−99B), and three small size (<10B). Regarding architectures, we have nine dense transformer and four hybrid transformers with 15 using Mixture-of-Experts. All the models have claimed their context length more than 128k tokens. Some of them have built-in reasoning switch to turn on and off thinking mode.

Table 1: Summary of the selected models in RULERV2.

| Model Name | Size | Claimed Length | Thinking | Huggingface / API |
|---|---|---|---|---|
| GPT5 (OpenAI, 2025b) | - | 400k | Y | gpt-5 (2025-08-07) |
| GPT4.1 (OpenAI, 2025a) | - | 1m | N | gpt-4.1 (2025-04-14) |
| o3 (OpenAI, 2025c) | - | 200k | Y | o3 (2025-04-16) |
| Gemini2.5 pro (Comanici et al., 2025) | - | 1m | Y | gemini-2.5-pro |
| Gemini2.5 flash (Comanici et al., 2025) | - | 1m | Y/N | gemini-2.5-flash |
| Claude4 Sonnet (Anthropic, 2025) | - | 200k | Y/N | claude-sonnet-4-20250514 |
| Grok4 (XAI, 2025) | - | 256k | Y | grok-4-0709 |
| Qwen3 235B Thinking 2507 (Qwen, 2025) | 235B-22B | 1m | Y | Qwen/Qwen3-235B-A22B-Thinking-2507 |
| Qwen3 235B Instruct 2507 (Qwen, 2025) | 235B-22B | 1m | N | Qwen/Qwen3-235B-A22B-Instruct-2507 |
| Qwen3 30B Thinking 2507 (Qwen, 2025) | 30B-3B | 1m | Y | Qwen/Qwen3-30B-A3B-Thinking-2507 |
| Qwen3 30B Instruct 2507 (Qwen, 2025) | 30B-3B | 1m | N | Qwen/Qwen3-30B-A3B-Instruct-2507 |
| Qwen3 80B Thinking Next (Qwen, 2025) | 80B-3B | 1m | Y | Qwen/Qwen3-Next-80B-A3B-Thinking |
| Qwen3 80B Instruct Next (Qwen, 2025) | 80B-3B | 1m | N | Qwen/Qwen3-Next-80B-A3B-Instruct |
| Qwen3 32B (Qwen, 2025) | 32.8B | 128k | Y/N | Qwen/Qwen3-32B |
| Qwen3 14B (Qwen, 2025) | 14.8B | 128k | Y/N | Qwen/Qwen3-14B |
| Qwen3 8B (Qwen, 2025) | 8.2B | 128k | Y/N | Qwen/Qwen3-8B |
| Qwen2.5 14B 1M (Yang et al., 2025) | 14.7B | 1m | N | Qwen/Qwen2.5-14B-Instruct-1M |
| Qwen2.5 7B 1M (Yang et al., 2025) | 7.6B | 1m | N | Qwen/Qwen2.5-7B-Instruct-1M |
| DeepSeek V3.1 (DeepSeek-AI, 2024) | 671B-37B | 128k | Y/N | deepseek-ai/DeepSeek-V3.1 |
| DeepSeek R1 0528 (DeepSeek-AI, 2025) | 671B-37B | 128k | Y | deepseek-ai/DeepSeek-R1-0528 |
| DeepSeek V3 0324 (DeepSeek-AI, 2024) | 671B-37B | 128k | N | deepseek-ai/DeepSeek-V3-0324 |
| Llama4 Maverick (Meta, 2025) | 400B-17B | 1m | N | meta-llama/Llama-4-Maverick-17B-128E-Instruct-FP8 |
| Llama4 Scout (Meta, 2025) | 109B-17B | 10m | N | meta-llama/Llama-4-Scout-17B-16E-Instruct |
| Llama3.1 70B (Dubey et al., 2024) | 70B | 128k | N | meta-llama/Llama-3.1-70B-Instruct |
| Llama3.1 8B (Dubey et al., 2024) | 8B | 128k | N | meta-llama/Llama-3.1-8B-Instruct |
| GPT OSS 120B (Agarwal et al., 2025) | 117B-5.1B | 128k | Y | openai/gpt-oss-120b |
| GPT OSS 20B (Agarwal et al., 2025) | 21B-3.6B | 128k | Y | openai/gpt-oss-20b |
| MiniMax M1-40k (Chen et al., 2025) | 456B-45.9B | 1m | Y | MiniMaxAI/MiniMax-M1-40k |
| MiniMax Text01 (Li et al., 2025a) | 456B-45.9B | 1m | N | MiniMaxAI/MiniMax-Text-01 |
| Seed OSS (ByteDanceSeed, 2025) | 36B | 512k | Y/N | ByteDance-Seed/Seed-OSS-36B-Instruct |
| GLM 4.5 (Zeng et al., 2025) | 355B-32B | 128k | Y/N | zai-org/GLM-4.5 |
| Nemotron1.5 Super (Bercovich et al., 2025) | 49.9B | 128k | Y/N | nvidia/Llama-3_3-Nemotron-Super-49B-v1_5 |
| Jamba1.7 Large (Lenz et al., 2024) | 398B-94B | 256k | N | ai21labs/AI21-Jamba-Large-1.7 |

# B    TASK TEMPLATES

The detailed task templates we used are provided in Tables 2, 3, and 4. We used MMLU (Hendrycks et al., 2020) as the base task for our multi-key and multi-value NIAH tasks and used HotPotQA (Yang et al., 2018) for the multi-doc QA task.

Table 2: Multi-key NIAH templates from basic to hard difficulties.

| | |
|---|---|
| Multi-key NIAH (Basic) Synthetic Retrieval | **Prompt:**
A special magic number is hidden within the following text. Make sure to memorize it. I will quiz you about the number afterwards.
One of the special magic numbers for word-1 is: number-1.
One of the special magic numbers for word-2 is: number-2.
......
One of the special magic numbers for word-x is: number-x.
......
One of the special magic numbers for word-n-1 is: number-n-1.
One of the special magic numbers for word-n is: number-n.

What is the special magic number for word-x mentioned in the provided text? The special magic number for word-x mentioned in the provided text is

**Expected Answer:** number-x |
| Multi-key NIAH (Easy) Realistic Retrieval | **Prompt:**
Below are some questions. I will ask you to copy one of them. Please copy and paste the question you find.
Question index-1: question-1.
Question index-2: question-2.
......
Question index-x: question-x.
......
Question index-n-1: question-n-1.
Question index-n: question-n.

Please copy the Question index-x from the context.

**Expected Answer:** question-x |
| Multi-key NIAH (Medium) Retrieve-then -solve | **Prompt:**
Below are some questions. I will ask you to solve one of them. Please solve the question you find and make sure to put the answer (and only answer) inside \boxed{}.
Question index-1: question-1.
Question index-2: question-2.
......
Question index-x: question-x.
......
Question index-n-1: question-n-1.
Question index-n: question-n.

Here are some examples to help you understand the task:
($\times N$)
Please copy the Question index from the context and then solve it with an answer from A, B, C, D.
Question index: question.
Solution: \boxed{answer}.

Here is the actual task you need to solve:
Please copy the Question index-x from the context and then solve it with an answer from A, B, C, D.

**Expected Answer:** \boxed{answer-x} |
| Multi-key NIAH (Hard) Single-step Solve | **Prompt:**
Below are some questions. I will ask you to solve one of them. Please solve the question you find and make sure to put the answer (and only answer) inside \boxed{}.
Question index-1: question-1.
Question index-2: question-2.
......
Question index-x: question-x.
......
Question index-n-1: question-n-1.
Question index-n: question-n.

Here are some examples to help you understand the task:
($\times N$)
Please solve the Question index from the context with an answer from A, B, C, D.
Solution: \boxed{answer}.

Here is the actual task you need to solve:
Please solve the Question index-x from the context with an answer from A, B, C, D.

**Expected Answer:** \boxed{answer-x} |

Table 3: Multi-value NIAH templates from basic to hard difficulties.

| | |
|---|---|
| Multi-value NIAH (Basic) Synthetic Retrieval | **Prompt:**
Some special magic numbers are hidden within the following text. Make sure to memorize them. I will quiz you about the numbers afterwards.
One of the special magic numbers for word-1 is: number-1.
......
One of the special magic numbers for word-x is: number-x1.
One of the special magic numbers for word-x is: number-x2.
One of the special magic numbers for word-x is: number-x3.
One of the special magic numbers for word-x is: number-x4.
......
One of the special magic numbers for word-n is: number-n.

What are all the special magic numbers for word-x mentioned in the provided text? The special magic numbers for word-x mentioned in the provided text are

**Expected Answer:** number-x1, number-x2, number-x3, number-x4 |
| Multi-value NIAH (Easy) Realistic Retrieval | **Prompt:**
Below are some questions. I will ask you to copy some of them. Please copy and paste the questions you find.
Question index-1: question-1.
Question index-2: question-2.
......
Question index-x: question-x1.
Question index-x: question-x2.
Question index-x: question-x3.
Question index-x: question-x4.
......
Question index-n-1: question-n-1.
Question index-n: question-n.

Please copy the Question index-x from the context.

**Expected Answer:** question-x1, question-x2, question-x3, question-x4 |
| Multi-value NIAH (Medium) Retrieve-then-solve | **Prompt:**
Below are some questions. I will ask you to copy one of them. Please copy and paste the question you find.
Question index-1: question-1.
Question index-2: question-2.
......
Question index-x: question-x1.
Question index-x: question-x2.
Question index-x: question-x3.
Question index-x: question-x4.
......
Question index-n-1: question-n-1.
Question index-n: question-n.

Please first copy all the Question index-x from the context and then copy the {order} (1 indexed) Question index-x at the end.

**Expected Answer:** question-x{order} |
| Multi-value NIAH (Hard) Single-step Solve | **Prompt:**
Below are some questions. I will ask you to copy one of them. Please copy and paste the question you find.
Question index-1: question-1.
Question index-2: question-2.
......
Question index-x: question-x1.
Question index-x: question-x2.
Question index-x: question-x3.
Question index-x: question-x4.
......
Question index-n-1: question-n-1.
Question index-n: question-n.

Please copy the {order} (1 indexed) Question index-x from the context.

**Expected Answer:** question-x{order} |

Table 4: Multi-doc QA templates from basic to hard difficulties.

| | |
|---|---|
| Multi-doc QA (Basic) Synthetic Retrieval | **Prompt:**
Below are some documents. I will give you a text at the end. Please find the document index of the text. Only give me the index without any document contents.
Document index-1: document-1.
Document index-2: document-2.
......
Document index-x: document-x.
......
Document index-n-1: document-n-1.
Document index-n: document-n.

Text: document-x
Most relevant document index:

**Expected Answer:** index-x |
| Multi-doc QA (Easy) Realistic Retrieval | **Prompt:**
Below are some documents. I will give you a question at the end. Please find the index of the most relevant document that can help answer the question. Only give me the index without any document contents.
Document index-1: document-1.
Document index-2: document-2.
......
Document index-x: document-x.
......
Document index-n-1: document-n-1.
Document index-n: document-n.

Question: question-x
Index of the most relevant document that can help answer the question:

**Expected Answer:** index-x |
| Multi-doc QA (Meidum) Retrieve-then-solve | **Prompt:**
Below are some documents. I will ask you to answer a question based on the documents. Please answer the question.
Document index-1: document-1.
Document index-2: document-2.
......
Document index-x: document-x.
......
Document index-n-1: document-n-1.
Document index-n: document-n.

Please first find and copy paste the documents relevant to the following question and then answer it based on the documents you find.
Question: question-x

**Expected Answer:** answer-x |
| Multi-doc QA (Hard) Single-step Solve | **Prompt:**
Below are some documents. I will ask you to answer a question based on the documents. Please answer the question.
Document index-1: document-1.
Document index-2: document-2.
......
Document index-x: document-x.
......
Document index-n-1: document-n-1.
Document index-n: document-n.

Please answer the following question based on the documents.
Question: question-x

**Expected Answer:** answer-x |

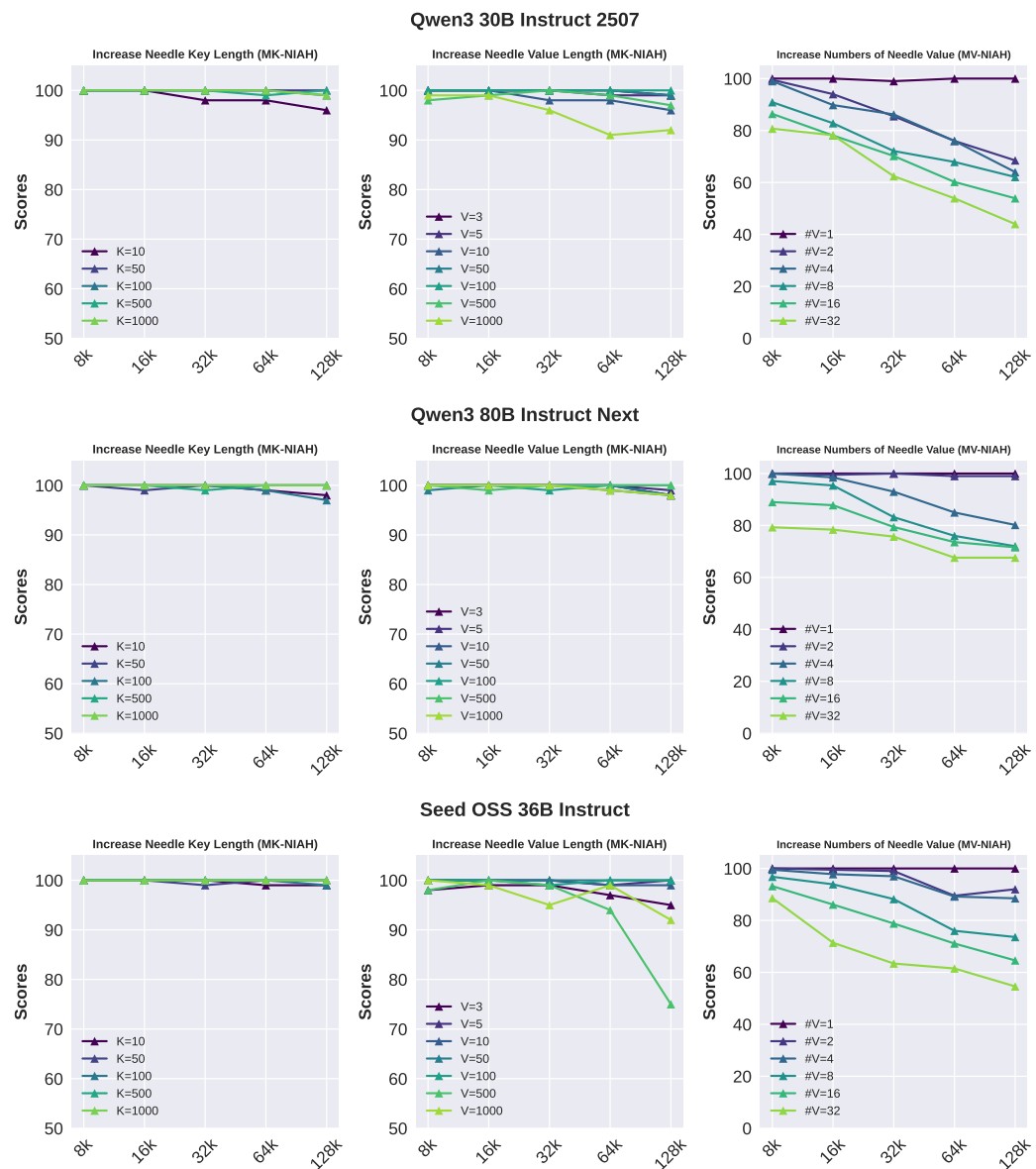

Figure 7: Additional results of Needle-in-a-haystack variants analysis.

## C ADDITIONAL ANALYSIS

### C.1 NEEDLE-IN-A-HAYSTACK VARIANTS

In Figure 7, we provide additional results from our analysis of needle-in-a-haystack variants. We found that most models show only slightly worse performance as the needle key length is short, but their scores degrade as the value length increases. Most importantly, all models consistently perform worse as the number of needle values grows, highlighting a significant challenge that existing LLMs struggle to reliably retrieve all relevant information, regardless of their architecture, family, or scale.

Table 5: RULERV2 full results (Part 1 of 5).

| Model | Sequence Length | Multi-key NIAH | | | | Multi-value NIAH | | | | Multi-doc QA | | | | Avg. ± 95% CI | Avg. 8K-128K |
|---|---|---|---|---|---|---|---|---|---|---|---|---|---|---|---|
| | | Basic | Easy | Medium | Hard | Basic | Easy | Medium | Hard | Basic | Easy | Medium | Hard | | |
| GPT5 Thinking high | 8192 | 100.0 | 100.0 | 92.0 | 96.0 | 100.0 | 100.0 | 100.0 | 98.1 | 100.0 | 95.0 | 84.1 | 88.3 | 96.1 ± 1.0 | 93.5 |
| | 16384 | 100.0 | 100.0 | 94.0 | 93.0 | 100.0 | 99.3 | 96.1 | 95.0 | 100.0 | 95.0 | 84.6 | 81.8 | 94.9 ± 1.2 | |
| | 32768 | 100.0 | 100.0 | 90.0 | 96.0 | 97.0 | 97.2 | 92.1 | 88.3 | 100.0 | 96.0 | 85.9 | 81.9 | 93.7 ± 1.3 | |
| | 65536 | 100.0 | 100.0 | 97.0 | 95.0 | 99.8 | 91.5 | 81.5 | 82.7 | 100.0 | 95.0 | 81.2 | 84.7 | 92.4 ± 1.4 | |
| | 110000 | 98.0 | 100.0 | 91.0 | 96.0 | 97.5 | 88.7 | 77.0 | 71.8 | 100.0 | 96.0 | 85.9 | 85.5 | 90.6 ± 1.5 | |
| Gemini2.5 Pro | 8192 | 100.0 | 100.0 | 85.0 | 90.0 | 99.2 | 99.4 | 98.1 | 99.1 | 100.0 | 93.0 | 83.0 | 86.0 | 94.4 ± 1.3 | 92.9 |
| | 16384 | 100.0 | 100.0 | 92.0 | 97.0 | 99.2 | 98.6 | 94.4 | 94.2 | 99.0 | 93.0 | 85.0 | 82.0 | 94.5 ± 1.2 | |
| | 32768 | 100.0 | 100.0 | 89.0 | 88.0 | 99.2 | 99.6 | 91.1 | 81.5 | 100.0 | 95.0 | 84.0 | 83.0 | 92.4 ± 1.5 | |
| | 65536 | 100.0 | 99.7 | 90.0 | 85.0 | 98.2 | 98.7 | 85.5 | 76.9 | 100.0 | 95.0 | 85.0 | 83.0 | 91.4 ± 1.6 | |
| | 110000 | 100.0 | 100.0 | 95.0 | 91.0 | 97.2 | 97.0 | 77.8 | 80.7 | 100.0 | 92.0 | 84.0 | 85.0 | 91.6 ± 1.5 | |
| Gemini2.5 Flash Thinking On | 8192 | 100.0 | 100.0 | 87.0 | 87.0 | 99.5 | 98.7 | 97.8 | 98.0 | 100.0 | 96.0 | 83.0 | 85.0 | 94.3 ± 1.3 | 91.4 |
| | 16384 | 100.0 | 99.1 | 89.0 | 95.0 | 98.0 | 97.8 | 96.1 | 89.3 | 99.0 | 95.0 | 83.0 | 83.0 | 93.7 ± 1.3 | |
| | 32768 | 100.0 | 100.0 | 88.0 | 84.0 | 97.0 | 98.6 | 91.0 | 77.9 | 99.0 | 91.0 | 85.0 | 85.0 | 91.4 ± 1.5 | |
| | 65536 | 99.0 | 100.0 | 87.0 | 79.0 | 94.8 | 96.7 | 81.7 | 67.0 | 99.0 | 95.0 | 80.0 | 82.0 | 88.4 ± 1.7 | |
| | 110000 | 99.0 | 100.0 | 94.0 | 86.0 | 91.2 | 94.0 | 80.5 | 66.0 | 100.0 | 94.0 | 81.0 | 82.5 | 89.0 ± 1.7 | |
| Gemini2.5 Flash Thinking Off | 8192 | 100.0 | 100.0 | 84.0 | 84.0 | 98.0 | 99.4 | 96.3 | 68.1 | 100.0 | 95.0 | 86.2 | 85.0 | 91.3 ± 1.6 | 88.0 |
| | 16384 | 100.0 | 100.0 | 80.0 | 90.0 | 91.0 | 98.4 | 94.1 | 54.7 | 100.0 | 94.0 | 84.0 | 82.0 | 89.0 ± 1.7 | |
| | 32768 | 100.0 | 99.1 | 87.0 | 86.0 | 88.8 | 97.9 | 92.5 | 53.7 | 100.0 | 95.0 | 87.0 | 81.0 | 88.8 ± 1.7 | |
| | 65536 | 100.0 | 100.0 | 89.0 | 79.0 | 77.8 | 96.5 | 82.6 | 43.2 | 100.0 | 91.0 | 85.0 | 80.0 | 85.5 ± 1.9 | |
| | 110000 | 98.0 | 99.0 | 85.0 | 82.0 | 76.0 | 91.3 | 77.0 | 55.0 | 100.0 | 92.0 | 82.0 | 77.0 | 85.5 ± 1.8 | |
| | 262144 | 98.0 | 97.1 | 87.0 | 77.0 | 70.5 | 84.8 | 62.8 | 41.1 | 99.0 | 95.0 | 84.0 | 77.0 | 82.5 ± 2.0 | |
| | 524288 | 100.0 | 95.0 | 90.0 | 90.0 | 62.5 | 76.6 | 50.1 | 31.2 | 95.0 | 94.0 | 90.0 | 70.0 | 79.1 ± 2.1 | |
| | 1000000 | | | | | | | 38.0 | 31.4 | | 85.0 | | | 77.0 ± 4.8 | |
| Claude4 Sonnet Thinking On | 8192 | 100.0 | 99.6 | 93.0 | 92.0 | 100.0 | 92.9 | 92.1 | 89.7 | 100.0 | 99.0 | 84.0 | 91.0 | 94.4 ± 1.2 | 90.8 |
| | 16384 | 100.0 | 99.7 | 86.0 | 91.0 | 99.8 | 92.5 | 77.9 | 80.1 | 100.0 | 97.0 | 84.0 | 90.0 | 91.5 ± 1.5 | |
| | 32768 | 100.0 | 99.6 | 92.0 | 93.0 | 99.5 | 90.2 | 77.8 | 72.9 | 100.0 | 98.0 | 85.0 | 90.0 | 91.5 ± 1.5 | |
| | 65536 | 100.0 | 99.6 | 96.0 | 93.0 | 99.2 | 88.0 | 74.5 | 65.8 | 100.0 | 99.0 | 82.0 | 89.0 | 90.5 ± 1.6 | |
| | 110000 | 100.0 | 97.2 | 90.0 | 93.0 | 97.2 | 77.4 | 59.9 | 47.4 | 100.0 | 97.0 | 83.0 | 89.0 | 85.9 ± 1.8 | |
| Claude4 Sonnet Thinking Off | 8192 | 100.0 | 99.6 | 92.0 | 92.0 | 100.0 | 92.7 | 87.2 | 55.8 | 100.0 | 100.0 | 86.2 | 89.0 | 91.2 ± 1.5 | 88.3 |
| | 16384 | 100.0 | 99.7 | 92.0 | 89.0 | 98.8 | 91.0 | 79.7 | 42.4 | 100.0 | 98.0 | 85.2 | 90.0 | 88.8 ± 1.7 | |
| | 32768 | 100.0 | 97.7 | 93.0 | 94.0 | 99.2 | 91.4 | 74.2 | 48.4 | 100.0 | 98.0 | 85.0 | 92.0 | 89.4 ± 1.7 | |
| | 65536 | 99.0 | 99.6 | 94.0 | 86.0 | 98.2 | 87.5 | 74.0 | 39.4 | 100.0 | 98.0 | 83.0 | 92.0 | 87.6 ± 1.8 | |
| | 110000 | 100.0 | 98.4 | 87.0 | 92.0 | 96.0 | 79.1 | 64.6 | 25.4 | 100.0 | 97.0 | 83.0 | 89.0 | 84.3 ± 1.9 | |
| GPT4.1 | 8192 | 100.0 | 100.0 | 86.0 | 87.0 | 100.0 | 99.4 | 96.2 | 58.5 | 100.0 | 98.0 | 85.0 | 84.0 | 91.2 ± 1.6 | 89.2 |
| | 16384 | 100.0 | 100.0 | 90.0 | 81.0 | 99.5 | 99.5 | 95.0 | 63.0 | 100.0 | 95.0 | 84.0 | 82.0 | 90.8 ± 1.7 | |
| | 32768 | 100.0 | 100.0 | 83.0 | 86.0 | 98.0 | 98.5 | 87.4 | 62.2 | 100.0 | 98.0 | 81.0 | 83.0 | 89.8 ± 1.7 | |
| | 65536 | 100.0 | 99.0 | 81.0 | 81.0 | 96.0 | 97.5 | 86.6 | 51.7 | 100.0 | 97.0 | 81.0 | 82.0 | 87.7 ± 1.8 | |
| | 110000 | 99.0 | 99.0 | 82.0 | 86.0 | 89.5 | 95.0 | 74.2 | 54.7 | 100.0 | 97.0 | 82.0 | 80.0 | 86.5 ± 1.8 | |
| | 262144 | 99.0 | 100.0 | 78.0 | 74.0 | 80.0 | 83.9 | 55.8 | 46.0 | 99.0 | 89.0 | 81.0 | 81.0 | 80.6 ± 2.1 | |
| | 524288 | 97.0 | 96.6 | 86.0 | 66.0 | 69.0 | 74.6 | 42.3 | 37.3 | 96.0 | 78.0 | 73.0 | 78.0 | 74.5 ± 2.3 | |
| | 1000000 | 90.0 | 95.2 | 80.0 | 65.0 | 61.3 | 75.0 | 67.9 | 47.7 | 100.0 | 65.0 | 85.0 | 70.0 | 75.2 ± 4.9 | |
| o3 Thinking high | 8192 | 98.0 | 99.9 | 92.0 | 96.0 | 97.5 | 94.4 | 97.0 | 86.2 | 100.0 | 97.0 | 80.0 | 81.5 | 93.3 ± 1.4 | 87.5 |
| | 16384 | 97.0 | 99.6 | 90.0 | 93.0 | 98.5 | 89.5 | 88.2 | 79.1 | 100.0 | 97.0 | 80.5 | 80.5 | 91.1 ± 1.5 | |
| | 32768 | 98.0 | 99.7 | 91.0 | 91.0 | 96.0 | 87.7 | 80.8 | 82.2 | 100.0 | 97.0 | 84.0 | 82.1 | 90.8 ± 1.5 | |
| | 65536 | 95.0 | 96.9 | 92.0 | 88.0 | 96.2 | 76.1 | 70.4 | 68.1 | 99.0 | 96.0 | 84.0 | 82.0 | 87.0 ± 1.8 | |
| | 110000 | 88.0 | 79.1 | 70.0 | 73.0 | 94.8 | 57.6 | 51.3 | 50.6 | 93.0 | 96.0 | 74.0 | 74.0 | 75.1 ± 2.3 | |

Table 6: RULERV2 full results (Part 2 of 5)

| Model | Sequence Length | Multi-key NIAH Basic | Easy | Medium | Hard | Multi-value NIAH Basic | Easy | Medium | Hard | Multi-doc QA Basic | Easy | Medium | Hard | Avg. ± 95% CI | Avg. 8K-128K |
|---|---|---|---|---|---|---|---|---|---|---|---|---|---|---|---|
| Qwen3 235B Thinking 2507 | 8192 | 100.0 | 98.0 | 90.0 | 82.0 | 98.2 | 90.6 | 92.0 | 84.5 | 100.0 | 97.0 | 87.0 | 96.0 | 92.9 ± 1.4 | 85.2 |
| | 16384 | 100.0 | 99.3 | 79.0 | 89.0 | 90.2 | 83.5 | 88.2 | 85.7 | 99.0 | 98.0 | 88.0 | 96.0 | 91.3 ± 1.4 | |
| | 32768 | 100.0 | 97.6 | 85.0 | 86.0 | 80.8 | 71.7 | 68.0 | 64.2 | 100.0 | 95.0 | 82.0 | 93.0 | 85.3 ± 1.8 | |
| | 65536 | 100.0 | 96.8 | 89.0 | 84.0 | 69.5 | 54.8 | 45.7 | 53.9 | 100.0 | 92.0 | 87.0 | 94.0 | 80.6 ± 2.0 | |
| | 110000 | 100.0 | 93.9 | 85.0 | 85.0 | 63.2 | 44.1 | 35.5 | 40.0 | 99.0 | 89.0 | 83.0 | 91.0 | 75.7 ± 2.2 | |
| Qwen3 235B Instruct 2507 | 8192 | 100.0 | 99.0 | 87.0 | 88.0 | 99.8 | 96.8 | 78.0 | 41.1 | 100.0 | 96.0 | 82.0 | 79.5 | 87.3 ± 1.8 | 83.7 |
| | 16384 | 100.0 | 98.0 | 88.0 | 85.0 | 98.5 | 96.8 | 60.0 | 49.7 | 100.0 | 91.0 | 81.0 | 81.0 | 85.8 ± 1.9 | |
| | 32768 | 100.0 | 99.1 | 87.0 | 87.0 | 97.0 | 92.4 | 58.2 | 36.7 | 100.0 | 94.0 | 79.0 | 84.0 | 84.5 ± 2.0 | |
| | 65536 | 100.0 | 97.2 | 92.0 | 83.0 | 93.8 | 87.7 | 59.8 | 32.1 | 98.0 | 83.0 | 80.0 | 79.0 | 82.5 ± 2.1 | |
| | 110000 | 100.0 | 77.9 | 90.0 | 81.0 | 89.2 | 77.4 | 45.3 | 27.1 | 89.0 | 78.0 | 76.0 | 79.0 | 78.2 ± 2.2 | |
| | 262144 | 99.0 | 50.6 | 63.0 | 70.0 | 68.2 | 49.8 | 35.2 | 25.9 | 78.0 | 68.0 | 70.0 | 68.0 | 65.3 ± 2.6 | |
| | 524288 | 92.0 | 22.6 | 57.0 | 65.0 | 52.5 | 28.4 | 16.1 | 16.9 | 40.0 | 44.0 | 66.0 | 70.0 | 53.0 ± 2.6 | |
| | 1000000 | 85.0 | – | 40.0 | 35.0 | 36.2 | 12.0 | 2.8 | 5.1 | – | 20.0 | 60.0 | 75.0 | 36.1 ± 5.6 | |
| Seed OSS Thinking On | 8192 | 100.0 | 99.6 | 85.0 | 93.0 | 95.8 | 86.4 | 86.4 | 81.6 | 100.0 | 100.0 | 81.2 | 77.9 | 90.6 ± 1.4 | 82.0 |
| | 16384 | 100.0 | 100.0 | 90.0 | 90.0 | 88.2 | 74.0 | 75.7 | 78.7 | 97.0 | 98.0 | 77.9 | 77.8 | 87.3 ± 1.6 | |
| | 32768 | 100.0 | 97.7 | 88.0 | 86.0 | 78.2 | 70.1 | 55.1 | 50.0 | 99.0 | 97.0 | 81.6 | 76.7 | 81.6 ± 1.9 | |
| | 65536 | 100.0 | 93.8 | 82.0 | 82.0 | 69.2 | 52.2 | 48.9 | 41.2 | 96.0 | 98.0 | 76.9 | 79.7 | 76.7 ± 2.1 | |
| | 110000 | 99.0 | 91.3 | 82.0 | 82.0 | 68.0 | 44.2 | 41.8 | 29.1 | 98.0 | 93.0 | 81.2 | 78.8 | 74.0 ± 2.2 | |
| Seed OSS Thinking Off | 8192 | 100.0 | 99.9 | 78.0 | 84.0 | 99.5 | 93.9 | 73.4 | 37.1 | 100.0 | 96.0 | 76.8 | 77.2 | 84.7 ± 2.0 | 79.2 |
| | 16384 | 100.0 | 100.0 | 79.0 | 80.0 | 97.8 | 85.5 | 63.5 | 37.7 | 98.0 | 95.0 | 76.8 | 76.9 | 82.5 ± 2.0 | |
| | 32768 | 100.0 | 98.7 | 83.0 | 87.0 | 97.2 | 78.4 | 49.1 | 25.5 | 100.0 | 92.0 | 77.3 | 72.4 | 80.1 ± 2.1 | |
| | 65536 | 100.0 | 92.2 | 75.0 | 77.0 | 90.2 | 62.6 | 42.4 | 28.0 | 100.0 | 96.0 | 76.7 | 75.1 | 76.3 ± 2.2 | |
| | 110000 | 100.0 | 88.8 | 75.0 | 83.0 | 88.2 | 55.8 | 25.6 | 18.9 | 96.0 | 93.0 | 70.1 | 74.4 | 72.4 ± 2.3 | |
| Grok4 Thinking On | 8192 | 100.0 | 100.0 | 90.0 | 94.0 | 99.8 | 96.0 | 88.5 | 87.1 | 100.0 | 96.0 | 88.0 | 80.9 | 93.4 ± 1.4 | 81.0 |
| | 16384 | 100.0 | 99.0 | 88.0 | 92.0 | 97.8 | 85.9 | 82.1 | 80.5 | 100.0 | 94.0 | 90.0 | 82.0 | 90.9 ± 1.5 | |
| | 32768 | 100.0 | 97.4 | 87.0 | 92.0 | 92.2 | 68.4 | 64.2 | 58.4 | 99.0 | 96.0 | 88.0 | 84.0 | 85.6 ± 1.8 | |
| | 65536 | 99.0 | 80.5 | 90.0 | 87.0 | 66.5 | 38.1 | 34.9 | 28.3 | 98.0 | 91.0 | 86.0 | 83.3 | 73.6 ± 2.3 | |
| | 110000 | 96.0 | 60.5 | 72.0 | 72.0 | 42.0 | 24.2 | 30.5 | 25.1 | 86.0 | 75.0 | 79.0 | 78.7 | 61.8 ± 2.5 | |
| Qwen3 30B Thinking 2507 | 8192 | 100.0 | 99.2 | 85.0 | 84.0 | 91.0 | 86.2 | 91.3 | 49.9 | 99.0 | 94.0 | 89.1 | 92.3 | 88.4 ± 1.7 | 79.7 |
| | 16384 | 100.0 | 98.6 | 85.0 | 92.0 | 79.0 | 76.6 | 58.2 | 33.5 | 99.0 | 95.0 | 91.0 | 93.0 | 83.4 ± 1.9 | |
| | 32768 | 100.0 | 98.7 | 84.0 | 85.0 | 73.2 | 65.4 | 51.5 | 28.3 | 98.0 | 93.0 | 85.1 | 92.0 | 79.5 ± 2.0 | |
| | 65536 | 100.0 | 99.8 | 84.0 | 85.0 | 62.7 | 47.6 | 38.3 | 26.3 | 98.0 | 93.0 | 83.0 | 84.8 | 75.2 ± 2.2 | |
| | 110000 | 99.0 | 97.7 | 89.0 | 83.0 | 57.0 | 35.5 | 26.6 | 18.0 | 98.0 | 89.0 | 83.1 | 84.6 | 71.7 ± 2.3 | |
| Qwen3 80B Thinking Next | 8192 | 100.0 | 98.9 | 89.0 | 89.0 | 95.5 | 86.4 | 83.5 | 85.8 | 100.0 | 92.0 | 76.2 | 86.3 | 90.2 ± 1.6 | 78.7 |
| | 16384 | 100.0 | 97.3 | 84.0 | 90.0 | 82.5 | 72.0 | 62.5 | 66.6 | 97.0 | 91.0 | 83.3 | 83.6 | 84.2 ± 1.9 | |
| | 32768 | 99.0 | 99.9 | 90.0 | 91.0 | 67.5 | 53.8 | 50.2 | 42.4 | 98.0 | 92.0 | 81.5 | 81.9 | 78.9 ± 2.0 | |
| | 65536 | 98.0 | 88.9 | 87.0 | 87.0 | 60.0 | 43.4 | 35.4 | 31.4 | 91.0 | 89.0 | 77.1 | 71.6 | 71.7 ± 2.3 | |
| | 110000 | 99.0 | 89.2 | 81.0 | 77.0 | 55.2 | 33.6 | 29.7 | 25.0 | 89.0 | 90.0 | 74.8 | 82.0 | 68.8 ± 2.4 | |
| DeepSeek V3 0324 | 8192 | 100.0 | 92.8 | 89.0 | 77.0 | 99.5 | 77.8 | 83.1 | 39.1 | 99.0 | 94.0 | 83.0 | 80.5 | 84.6 ± 1.4 | 77.4 |
| | 16384 | 100.0 | 90.4 | 74.0 | 80.0 | 92.8 | 79.7 | 71.2 | 34.7 | 100.0 | 92.0 | 77.0 | 81.0 | 81.1 ± 2.1 | |
| | 32768 | 100.0 | 90.8 | 83.0 | 78.0 | 86.8 | 80.0 | 67.9 | 15.5 | 99.0 | 88.0 | 76.0 | 80.0 | 78.8 ± 2.2 | |
| | 65536 | 100.0 | 88.1 | 79.0 | 79.0 | 83.2 | 67.0 | 50.9 | 7.7 | 96.0 | 89.0 | 75.0 | 76.0 | 74.2 ± 2.3 | |
| | 110000 | 99.0 | 82.0 | 78.0 | 73.0 | 77.0 | 58.3 | 37.2 | 6.2 | 89.0 | 83.0 | 71.0 | 68.0 | 68.5 ± 2.4 | |

Table 7: RULERV2 full results (Part 3 of 5)

| Model | Sequence Length | Multi-key NIAH | | | | Multi-value NIAH | | | | Multi-doc QA | | | | Avg. ± 95% CI | Avg. 8K-128K |
|---|---|---|---|---|---|---|---|---|---|---|---|---|---|---|---|
| | | Basic | Easy | Medium | Hard | Basic | Easy | Medium | Hard | Basic | Easy | Medium | Hard | | |
| DeepSeek V3.1 Think On | 8192 | 100.0 | 100.0 | 90.0 | 95.0 | 96.8 | 85.4 | 65.4 | 27.7 | 100.0 | 96.0 | 82.3 | 75.4 | 84.5 ± 1.9 | 76.9 |
| | 16384 | 100.0 | 99.1 | 91.0 | 94.0 | 81.0 | 77.1 | 56.4 | 17.9 | 100.0 | 96.0 | 76.9 | 74.6 | 80.3 ± 2.0 | |
| | 32768 | 100.0 | 100.0 | 87.0 | 95.0 | 72.8 | 67.1 | 53.3 | 10.7 | 100.0 | 95.0 | 81.2 | 76.6 | 78.2 ± 2.1 | |
| | 65536 | 99.0 | 95.0 | 93.0 | 92.0 | 62.5 | 55.0 | 31.8 | 4.4 | 97.0 | 97.0 | 72.8 | 76.3 | 73.0 ± 2.3 | |
| | 110000 | 100.0 | 86.6 | 85.0 | 83.0 | 59.0 | 37.8 | 31.1 | 5.7 | 96.0 | 91.0 | 74.2 | 70.0 | 68.3 ± 2.4 | |
| DeepSeek V3.1 Think Off | 8192 | 100.0 | 100.0 | 85.0 | 84.0 | 98.5 | 93.6 | 43.4 | 17.9 | 100.0 | 86.0 | 81.5 | 73.5 | 80.3 ± 2.2 | 73.9 |
| | 16384 | 100.0 | 98.1 | 86.0 | 85.0 | 92.2 | 87.8 | 49.0 | 22.1 | 100.0 | 81.0 | 82.2 | 75.2 | 79.9 ± 2.1 | |
| | 32768 | 100.0 | 99.9 | 81.0 | 88.0 | 83.0 | 74.4 | 39.3 | 27.0 | 100.0 | 70.0 | 77.3 | 76.9 | 76.4 ± 2.2 | |
| | 65536 | 100.0 | 96.2 | 85.0 | 84.0 | 67.0 | 47.7 | 31.5 | 16.9 | 97.0 | 50.0 | 74.0 | 67.9 | 68.1 ± 2.4 | |
| | 110000 | 100.0 | 95.3 | 84.0 | 83.0 | 62.7 | 27.0 | 29.7 | 21.2 | 94.0 | 40.0 | 70.0 | 70.7 | 64.8 ± 2.5 | |
| GLM4.5 Think On | 8192 | 100.0 | 98.9 | 88.0 | 84.0 | 91.5 | 83.6 | 87.3 | 76.9 | 97.0 | 97.0 | 80.0 | 81.0 | 88.8 ± 1.6 | 75.9 |
| | 16384 | 100.0 | 97.7 | 85.0 | 80.0 | 88.0 | 77.0 | 61.9 | 42.3 | 100.0 | 96.0 | 80.0 | 88.0 | 83.0 ± 2.0 | |
| | 32768 | 100.0 | 89.3 | 92.0 | 84.0 | 91.2 | 61.2 | 49.8 | 40.1 | 99.0 | 96.0 | 77.0 | 86.0 | 80.5 ± 2.1 | |
| | 65536 | 100.0 | 78.5 | 87.0 | 76.0 | 86.5 | 34.6 | 30.3 | 25.3 | 98.0 | 94.0 | 83.0 | 80.0 | 72.8 ± 2.4 | |
| | 110000 | 94.0 | 27.2 | 46.0 | 45.0 | 79.2 | 14.4 | 13.6 | 9.4 | 86.0 | 93.0 | 73.0 | 76.0 | 54.7 ± 2.7 | |
| GLM4.5 Think Off | 8192 | 100.0 | 100.0 | 83.0 | 86.0 | 99.0 | 93.0 | 60.6 | 34.4 | 100.0 | 95.0 | 80.0 | 84.0 | 84.6 ± 2.0 | 74.9 |
| | 16384 | 100.0 | 95.1 | 84.0 | 81.0 | 97.2 | 83.8 | 51.3 | 40.3 | 100.0 | 94.0 | 81.0 | 86.0 | 83.3 ± 2.0 | |
| | 32768 | 100.0 | 79.4 | 80.0 | 89.0 | 94.8 | 70.6 | 46.6 | 37.3 | 99.0 | 87.0 | 77.0 | 81.0 | 80.5 ± 2.1 | |
| | 65536 | 100.0 | 79.4 | 77.0 | 68.0 | 87.8 | 50.7 | 28.6 | 18.5 | 99.0 | 80.0 | 74.0 | 79.0 | 70.8 ± 2.4 | |
| | 110000 | 92.0 | 32.3 | 51.0 | 47.0 | 68.2 | 20.2 | 14.3 | 16.6 | 97.0 | 80.0 | 69.0 | 76.0 | 55.3 ± 2.6 | |
| DeepSeek R1 0528 | 8192 | 100.0 | 98.8 | 87.0 | 92.0 | 94.2 | 87.9 | 73.0 | 43.4 | 100.0 | 96.0 | 81.0 | 79.8 | 86.1 ± 1.8 | 75.2 |
| | 16384 | 100.0 | 95.2 | 92.0 | 89.0 | 91.2 | 81.1 | 62.7 | 25.0 | 100.0 | 98.0 | 83.0 | 77.6 | 82.9 ± 2.0 | |
| | 32768 | 99.0 | 91.6 | 88.0 | 89.0 | 81.2 | 59.2 | 51.2 | 15.1 | 99.0 | 97.0 | 78.0 | 75.7 | 77.0 ± 2.1 | |
| | 65536 | 100.0 | 81.5 | 79.0 | 78.0 | 73.8 | 34.3 | 35.6 | 11.4 | 92.0 | 85.0 | 76.0 | 75.2 | 68.5 ± 2.4 | |
| | 110000 | 98.0 | 67.0 | 77.0 | 74.0 | 61.8 | 22.7 | 29.4 | 3.4 | 84.0 | 71.0 | 73.0 | 77.3 | 61.6 ± 2.6 | |
| Qwen3 30B Instruct 2507 | 8192 | 100.0 | 98.8 | 81.0 | 73.0 | 98.8 | 93.4 | 32.8 | 28.0 | 98.0 | 83.0 | 73.0 | 78.0 | 78.2 ± 2.3 | 74.6 |
| | 16384 | 100.0 | 98.8 | 76.0 | 69.0 | 89.2 | 90.6 | 28.9 | 29.8 | 93.0 | 80.0 | 77.0 | 80.0 | 76.0 ± 2.3 | |
| | 32768 | 99.0 | 100.0 | 84.0 | 82.0 | 85.5 | 80.6 | 30.6 | 29.8 | 89.0 | 77.0 | 69.0 | 80.0 | 75.5 ± 2.3 | |
| | 65536 | 100.0 | 100.0 | 82.0 | 74.0 | 77.2 | 72.5 | 30.5 | 31.0 | 92.0 | 63.0 | 81.0 | 77.0 | 73.4 ± 2.4 | |
| | 110000 | 100.0 | 99.9 | 83.0 | 78.0 | 66.8 | 49.0 | 33.4 | 31.4 | 90.0 | 55.0 | 73.0 | 79.0 | 69.9 ± 2.4 | |
| GPT OSS 120B Thinking high | 8192 | 99.0 | 98.9 | 90.0 | 78.0 | 99.8 | 64.7 | 73.3 | 89.6 | 100.0 | 98.0 | 93.0 | 76.0 | 88.4 ± 0.6 | 74.3 |
| | 16384 | 92.0 | 95.6 | 81.0 | 86.0 | 93.8 | 68.4 | 43.8 | 75.8 | 99.0 | 95.0 | 87.0 | 74.1 | 82.6 ± 0.8 | |
| | 32768 | 72.0 | 91.7 | 85.0 | 80.0 | 81.5 | 42.6 | 21.0 | 49.2 | 99.0 | 94.0 | 87.0 | 75.1 | 73.2 ± 1.0 | |
| | 65536 | 69.0 | 83.6 | 77.0 | 85.0 | 71.5 | 24.5 | 25.8 | 44.6 | 97.0 | 81.0 | 87.0 | 70.0 | 68.0 ± 1.0 | |
| | 110000 | 60.0 | 75.7 | 76.0 | 73.0 | 47.8 | 16.2 | 23.7 | 28.4 | 82.0 | 84.0 | 76.0 | 71.7 | 59.5 ± 1.2 | |
| MiniMax M1-40k | 8192 | 99.0 | 92.2 | 89.0 | 88.0 | 96.0 | 70.6 | 73.4 | 49.6 | 100.0 | 97.0 | 78.2 | 75.8 | 84.1 ± 1.9 | 72.8 |
| | 16384 | 95.0 | 91.2 | 86.0 | 87.0 | 88.5 | 60.2 | 63.7 | 34.3 | 99.0 | 98.0 | 83.0 | 75.1 | 79.9 ± 2.1 | |
| | 32768 | 83.0 | 84.2 | 82.0 | 74.0 | 78.0 | 57.9 | 49.7 | 28.3 | 99.0 | 97.0 | 77.8 | 75.3 | 73.9 ± 2.3 | |
| | 65536 | 61.0 | 65.3 | 75.0 | 81.0 | 65.0 | 48.5 | 39.6 | 13.3 | 98.0 | 95.0 | 75.0 | 73.7 | 65.9 ± 2.5 | |
| | 110000 | 55.0 | 58.0 | 70.0 | 71.0 | 52.0 | 42.0 | 26.8 | 20.0 | 94.0 | 90.0 | 76.0 | 70.4 | 60.4 ± 2.6 | |
| Qwen3 80B Instruct Next | 8192 | 100.0 | 99.0 | 89.0 | 91.0 | 99.8 | 91.0 | 49.2 | 41.7 | 85.0 | 55.0 | 79.0 | 84.0 | 80.3 ± 2.2 | 71.8 |
| | 16384 | 100.0 | 99.0 | 85.0 | 90.0 | 96.8 | 86.4 | 44.9 | 32.0 | 76.0 | 47.0 | 79.0 | 82.0 | 76.5 ± 2.3 | |
| | 32768 | 99.0 | 98.1 | 83.0 | 88.0 | 88.5 | 55.4 | 38.6 | 27.2 | 67.0 | 40.0 | 75.0 | 84.0 | 72.0 ± 2.4 | |
| | 65536 | 100.0 | 95.0 | 80.0 | 77.0 | 76.8 | 40.5 | 37.8 | 23.5 | 64.0 | 37.0 | 70.0 | 82.0 | 66.5 ± 2.5 | |
| | 110000 | 99.0 | 84.3 | 86.0 | 77.0 | 68.8 | 24.9 | 29.5 | 23.9 | 62.0 | 39.0 | 74.0 | 78.0 | 63.6 ± 2.5 | |
| | 262144 | 95.0 | 61.2 | 60.0 | 52.0 | 51.0 | 18.5 | 22.3 | 29.1 | 30.0 | 19.0 | 66.5 | 75.0 | 49.2 ± 2.6 | |
| | 524288 | 95.0 | 45.3 | 49.0 | 42.0 | 42.8 | 14.2 | 14.2 | 14.0 | 15.0 | 12.0 | 67.0 | 64.0 | 39.9 ± 2.6 | |
| | 1000000 | 40.0 | 18.0 | 20.0 | 35.0 | 25.0 | 11.6 | 16.2 | 18.0 | 0.0 | 5.0 | 52.0 | 52.5 | 24.4 ± 5.0 | |

Table 8: RULERV2 full results (Part 4 of 5)

| Model | Sequence Length | Multi-key NIAH | | | | Multi-value NIAH | | | | Multi-doc QA | | | | Avg. ± 95% CI | Avg. 8K-128K |
|---|---|---|---|---|---|---|---|---|---|---|---|---|---|---|---|
| | | Basic | Easy | Medium | Hard | Basic | Easy | Medium | Hard | Basic | Easy | Medium | Hard | | |
| Qwen 2.5 14B 1M | 8192 | 100.0 | 97.9 | 75.0 | 77.0 | 87.5 | 64.0 | 39.5 | 44.8 | 87.0 | 73.0 | 79.0 | 82.0 | 75.6 ± 2.2 | 68.0 |
| | 16384 | 100.0 | 99.6 | 77.0 | 71.0 | 74.0 | 63.6 | 39.0 | 31.2 | 86.0 | 65.0 | 71.0 | 80.0 | 71.5 ± 2.3 | |
| | 32768 | 100.0 | 96.2 | 72.0 | 74.0 | 60.2 | 55.4 | 30.7 | 37.9 | 82.0 | 50.0 | 69.0 | 79.0 | 67.2 ± 2.4 | |
| | 65536 | 99.0 | 94.5 | 71.0 | 73.0 | 56.0 | 45.6 | 37.4 | 28.3 | 71.0 | 39.0 | 68.0 | 74.0 | 63.1 ± 2.5 | |
| | 110000 | 100.0 | 98.4 | 76.0 | 76.0 | 53.2 | 42.5 | 32.5 | 33.4 | 50.0 | 46.0 | 75.0 | 67.0 | 62.5 ± 2.5 | |
| Qwen3 32B Thinking On | 8192 | 99.0 | 99.0 | 77.0 | 78.0 | 83.2 | 65.6 | 53.9 | 58.3 | 99.0 | 96.0 | 81.0 | 82.0 | 81.0 ± 2.0 | 67.8 |
| | 16384 | 99.0 | 88.0 | 78.0 | 73.0 | 75.0 | 48.5 | 38.9 | 34.0 | 93.0 | 86.0 | 82.0 | 82.0 | 73.1 ± 2.3 | |
| | 32768 | 98.0 | 64.1 | 72.0 | 75.0 | 75.2 | 43.3 | 26.6 | 29.9 | 85.0 | 84.0 | 80.0 | 80.0 | 67.8 ± 2.4 | |
| | 65536 | 92.0 | 71.4 | 69.0 | 66.0 | 69.2 | 30.4 | 28.8 | 21.0 | 82.0 | 69.0 | 73.0 | 77.0 | 62.4 ± 2.5 | |
| | 110000 | 81.0 | 60.9 | 61.0 | 54.0 | 65.8 | 23.9 | 21.9 | 10.3 | 72.0 | 52.0 | 75.0 | 78.0 | 54.7 ± 2.6 | |
| Qwen3 32B Thinking Off | 8192 | 100.0 | 95.9 | 77.0 | 68.0 | 87.0 | 76.1 | 46.6 | 32.8 | 89.0 | 73.0 | 82.0 | 79.0 | 75.5 ± 2.3 | 64.1 |
| | 16384 | 100.0 | 94.0 | 80.0 | 60.0 | 76.0 | 55.2 | 32.8 | 30.4 | 92.0 | 55.0 | 75.2 | 76.0 | 68.9 ± 2.5 | |
| | 32768 | 98.0 | 83.5 | 73.0 | 67.0 | 70.5 | 45.4 | 29.9 | 25.2 | 81.0 | 54.0 | 67.0 | 81.0 | 64.6 ± 2.5 | |
| | 65536 | 90.0 | 80.7 | 74.0 | 53.0 | 66.2 | 33.2 | 28.0 | 16.0 | 76.0 | 48.0 | 66.2 | 70.0 | 58.4 ± 2.6 | |
| | 110000 | 87.0 | 76.5 | 63.0 | 53.0 | 60.5 | 26.4 | 24.5 | 13.9 | 64.0 | 32.0 | 66.2 | 70.0 | 53.1 ± 2.6 | |
| Llama4 Maverick | 8192 | 95.0 | 99.6 | 87.0 | 86.0 | 96.8 | 83.2 | 58.2 | 36.9 | 100.0 | 97.0 | 81.0 | 88.0 | 84.1 ± 1.9 | 67.5 |
| | 16384 | 98.0 | 93.0 | 73.0 | 76.0 | 88.0 | 71.8 | 48.2 | 29.0 | 98.0 | 96.0 | 81.0 | 83.0 | 77.9 ± 2.1 | |
| | 32768 | 85.0 | 82.3 | 76.0 | 77.0 | 59.0 | 51.4 | 28.4 | 23.6 | 89.0 | 91.0 | 81.3 | 77.5 | 68.5 ± 2.3 | |
| | 65536 | 76.0 | 59.8 | 70.0 | 55.0 | 37.5 | 34.9 | 24.2 | 23.8 | 78.0 | 74.0 | 79.3 | 79.0 | 57.6 ± 2.5 | |
| | 110000 | 51.0 | 41.7 | 56.0 | 45.0 | 27.5 | 27.9 | 22.6 | 24.2 | 63.0 | 74.0 | 78.0 | 80.0 | 49.2 ± 2.6 | |
| | 262144 | 27.0 | 26.3 | 45.0 | 21.0 | 13.5 | 17.1 | 11.4 | 14.3 | 36.0 | 42.0 | 75.0 | 74.0 | 33.6 ± 2.3 | |
| | 524288 | 10.0 | 15.6 | 34.0 | 19.0 | 6.8 | 9.1 | 4.2 | 8.6 | 25.0 | 28.0 | 79.0 | 73.3 | 26.1 ± 2.4 | |
| | 1000000 | 15.0 | 13.9 | 25.0 | 15.0 | 8.8 | 5.1 | 3.1 | 2.4 | 5.0 | 20.0 | 77.5 | 80.0 | 22.6 ± 2.5 | |
| MiniMax Text 01 | 8192 | 100.0 | 88.5 | 89.0 | 81.0 | 70.8 | 14.8 | 30.9 | 43.7 | 99.0 | 76.0 | 76.0 | 73.8 | 70.3 ± 2.5 | 64.8 |
| | 16384 | 100.0 | 84.5 | 76.0 | 76.0 | 67.5 | 10.3 | 36.8 | 34.1 | 98.0 | 74.0 | 73.0 | 70.1 | 66.7 ± 2.4 | |
| | 32768 | 98.0 | 83.8 | 80.0 | 73.0 | 49.5 | 12.4 | 28.2 | 30.8 | 91.0 | 77.0 | 74.0 | 69.8 | 64.0 ± 2.5 | |
| | 65536 | 96.0 | 83.7 | 71.0 | 67.0 | 42.5 | 14.0 | 31.8 | 29.4 | 88.0 | 65.0 | 71.0 | 65.8 | 60.4 ± 2.5 | |
| | 110000 | 94.0 | 85.7 | 74.0 | 78.0 | 35.5 | 17.9 | 27.6 | 31.9 | 91.0 | 70.0 | 71.7 | 73.6 | 62.6 ± 5.7 | |
| | 262144 | 96.0 | 79.8 | 77.0 | 74.0 | 34.0 | 19.8 | 22.0 | 28.1 | 80.0 | 69.0 | 65.0 | 68.6 | 59.4 ± 0.7 | |
| | 524288 | 97.0 | 80.9 | 66.0 | 63.0 | 32.0 | 22.3 | 18.4 | 21.2 | 81.0 | 65.0 | 67.2 | 64.2 | 56.5 ± 1.0 | |
| | 1000000 | 90.0 | 59.4 | 70.0 | 45.0 | 26.2 | 25.0 | 19.0 | 21.9 | 45.0 | 50.0 | 35.0 | 65.0 | 46.0 ± 1.1 | |
| GPT OSS 20B Thinking high | 8192 | 100.0 | 99.6 | 85.0 | 81.0 | 92.2 | 43.4 | 75.7 | 70.0 | 99.0 | 97.0 | 90.0 | 75.0 | 84.0 ± 1.2 | 64.4 |
| | 16384 | 96.0 | 87.6 | 74.0 | 78.0 | 73.0 | 34.6 | 39.8 | 43.0 | 99.0 | 94.0 | 86.0 | 70.0 | 72.9 ± 1.2 | |
| | 32768 | 79.0 | 80.5 | 81.0 | 78.0 | 56.8 | 22.0 | 28.3 | 40.3 | 96.0 | 91.0 | 81.0 | 67.0 | 66.7 ± 2.0 | |
| | 65536 | 76.0 | 69.1 | 64.0 | 55.0 | 49.8 | 20.1 | 19.8 | 26.2 | 85.0 | 86.0 | 70.0 | 64.0 | 57.1 ± 2.2 | |
| | 110000 | 55.0 | 39.0 | 43.0 | 47.0 | 26.0 | 11.0 | 13.7 | 10.5 | 68.0 | 71.0 | 58.0 | 53.0 | 41.3 ± 2.4 | |
| Llama 3.1 70B | 8192 | 100.0 | 98.1 | 75.0 | 69.0 | 87.2 | 80.5 | 57.4 | 43.2 | 99.0 | 92.0 | 80.0 | 81.4 | 80.2 ± 2.4 | 63.7 |
| | 16384 | 100.0 | 89.6 | 68.0 | 63.0 | 80.5 | 67.0 | 53.9 | 35.4 | 100.0 | 90.0 | 75.7 | 76.8 | 75.0 ± 2.4 | |
| | 32768 | 100.0 | 86.4 | 65.0 | 62.0 | 70.2 | 58.2 | 34.5 | 27.6 | 99.0 | 84.0 | 74.2 | 67.7 | 69.1 ± 2.2 | |
| | 65536 | 97.0 | 79.3 | 62.0 | 50.0 | 61.3 | 45.1 | 27.8 | 23.3 | 93.0 | 59.0 | 67.3 | 68.5 | 61.1 ± 2.4 | |
| | 110000 | 51.0 | 50.0 | 33.0 | 16.0 | 31.2 | 14.0 | 12.7 | 14.1 | 45.0 | 33.0 | 49.2 | 50.4 | 33.3 ± 2.5 | |
| Qwen3 14B Thinking On | 8192 | 100.0 | 88.1 | 72.0 | 71.0 | 82.2 | 60.2 | 59.1 | 41.6 | 94.0 | 93.0 | 76.0 | 76.8 | 76.2 ± 2.6 | 62.0 |
| | 16384 | 99.0 | 81.2 | 61.0 | 62.0 | 73.5 | 44.8 | 35.1 | 30.3 | 93.0 | 85.0 | 80.0 | 81.2 | 68.8 ± 2.6 | |
| | 32768 | 95.0 | 65.6 | 66.0 | 60.0 | 66.5 | 36.8 | 29.5 | 18.2 | 82.0 | 74.0 | 73.0 | 80.0 | 62.2 ± 2.4 | |
| | 65536 | 91.0 | 68.1 | 55.0 | 46.0 | 68.5 | 21.0 | 19.6 | 11.0 | 76.0 | 65.0 | 67.0 | 74.5 | 55.2 ± 2.5 | |
| | 110000 | 93.0 | 46.6 | 40.0 | 48.0 | 58.5 | 19.7 | 19.0 | 8.1 | 55.0 | 52.0 | 62.0 | 68.5 | 47.5 ± 2.6 | |

Table 9: RULERV2 full results (Part 5 of 5)

| Model | Sequence Length | Multi-key NIAH Basic | Easy | Medium | Hard | Multi-value NIAH Basic | Easy | Medium | Hard | Multi-doc QA Basic | Easy | Medium | Hard | Avg. ± 95% CI | Avg. 8K-128K |
|---|---|---|---|---|---|---|---|---|---|---|---|---|---|---|---|
| Qwen3 14B Thinking Off | 8192 | 100.0 | 79.6 | 62.0 | 65.0 | 85.5 | 65.0 | 31.7 | 33.5 | 87.0 | 77.0 | 75.0 | 82.0 | 70.3 ± 2.6 | 57.2 |
| | 16384 | 100.0 | 75.4 | 65.0 | 64.0 | 77.8 | 49.9 | 27.9 | 24.7 | 79.0 | 77.0 | 71.0 | 81.0 | 66.1 ± 2.6 | |
| | 32768 | 95.0 | 66.0 | 61.0 | 61.0 | 60.5 | 29.9 | 29.4 | 23.2 | 62.0 | 59.0 | 53.0 | 75.0 | 56.3 ± 2.2 | |
| | 65536 | 95.0 | 61.7 | 57.0 | 42.0 | 57.0 | 23.3 | 14.0 | 17.5 | 55.0 | 42.0 | 59.0 | 71.0 | 49.5 ± 2.4 | |
| | 110000 | 89.0 | 45.7 | 47.0 | 43.0 | 52.5 | 16.0 | 9.5 | 13.4 | 60.0 | 25.0 | 54.0 | 68.0 | 43.6 ± 2.5 | |
| Llama4 Scout | 8192 | 100.0 | 97.3 | 78.0 | 76.0 | 91.0 | 62.1 | 29.6 | 24.2 | 98.0 | 87.0 | 89.0 | 86.0 | 76.5 ± 2.6 | 60.7 |
| | 16384 | 96.0 | 92.6 | 67.0 | 68.0 | 73.5 | 49.0 | 22.8 | 18.8 | 91.0 | 77.0 | 86.0 | 84.3 | 68.8 ± 2.6 | |
| | 32768 | 84.0 | 65.6 | 60.0 | 54.0 | 56.8 | 36.5 | 16.2 | 21.2 | 84.0 | 62.0 | 82.0 | 82.0 | 58.7 ± 5.7 | |
| | 65536 | 83.0 | 57.1 | 62.0 | 41.0 | 46.2 | 25.7 | 16.9 | 16.7 | 62.0 | 57.0 | 75.5 | 77.5 | 51.7 ± 5.5 | |
| | 110000 | 91.0 | 48.5 | 49.0 | 44.0 | 39.5 | 21.8 | 17.1 | 23.5 | 46.0 | 43.0 | 73.0 | 73.9 | 47.5 ± 5.2 | |
| Nemotron1.5 Super Thinking On | 8192 | 99.0 | 83.0 | 84.0 | 87.0 | 61.5 | 60.6 | 55.5 | 43.3 | 98.0 | 95.0 | 77.2 | 77.8 | 76.8 ± 2.1 | 57.2 |
| | 16384 | 99.0 | 82.4 | 75.0 | 79.0 | 51.7 | 49.5 | 31.6 | 22.5 | 99.0 | 94.0 | 76.2 | 73.5 | 69.5 ± 2.3 | |
| | 32768 | 92.0 | 62.0 | 49.0 | 65.0 | 44.0 | 31.6 | 26.9 | 14.3 | 96.0 | 89.0 | 77.0 | 74.2 | 60.1 ± 2.6 | |
| | 65536 | 78.0 | 26.5 | 41.0 | 42.0 | 38.2 | 19.8 | 16.9 | 8.4 | 87.0 | 75.0 | 69.9 | 71.1 | 47.8 ± 2.6 | |
| | 110000 | 42.0 | 12.0 | 33.0 | 39.0 | 20.8 | 10.8 | 5.3 | 6.0 | 55.0 | 45.0 | 58.2 | 55.6 | 31.9 ± 2.5 | |
| Nemotron1.5 Super Thinking Off | 8192 | 100.0 | 77.8 | 76.0 | 78.0 | 67.8 | 54.5 | 49.3 | 22.5 | 100.0 | 88.0 | 78.0 | 83.0 | 72.9 ± 2.2 | 55.2 |
| | 16384 | 99.0 | 73.1 | 72.0 | 62.0 | 50.2 | 39.0 | 38.4 | 12.9 | 95.0 | 80.0 | 74.0 | 84.0 | 65.0 ± 2.5 | |
| | 32768 | 97.0 | 64.3 | 66.0 | 46.0 | 38.2 | 14.2 | 21.4 | 11.7 | 94.0 | 67.0 | 75.0 | 77.0 | 56.0 ± 2.6 | |
| | 65536 | 78.0 | 37.9 | 63.0 | 39.0 | 36.8 | 13.7 | 15.8 | 12.1 | 88.0 | 64.0 | 65.0 | 68.0 | 48.4 ± 2.6 | |
| | 110000 | 41.0 | 20.7 | 31.0 | 29.0 | 22.2 | 5.2 | 5.6 | 10.0 | 66.0 | 47.0 | 63.3 | 62.0 | 33.6 ± 2.6 | |
| Qwen 2.5 7B 1M | 8192 | 100.0 | 98.1 | 62.0 | 68.0 | 55.0 | 51.5 | 38.7 | 32.1 | 70.0 | 21.0 | 74.2 | 78.0 | 62.4 ± 2.5 | 56.8 |
| | 16384 | 100.0 | 96.5 | 63.0 | 58.0 | 52.5 | 53.4 | 33.8 | 28.6 | 47.0 | 20.0 | 70.0 | 75.0 | 58.2 ± 2.5 | |
| | 32768 | 100.0 | 95.8 | 61.0 | 66.0 | 49.2 | 37.0 | 35.5 | 27.3 | 45.0 | 12.0 | 67.0 | 70.0 | 55.5 ± 2.6 | |
| | 65536 | 99.0 | 95.7 | 74.0 | 52.0 | 44.2 | 35.2 | 33.1 | 28.2 | 46.0 | 17.0 | 59.0 | 68.0 | 54.3 ± 2.6 | |
| | 110000 | 94.0 | 93.5 | 63.0 | 57.0 | 42.2 | 37.3 | 34.1 | 25.4 | 46.0 | 26.0 | 59.0 | 65.0 | 53.5 ± 2.6 | |
| Jamba1.7 Large | 8192 | 100.0 | 94.9 | 70.0 | 65.0 | 87.2 | 40.6 | 30.8 | 28.6 | 82.0 | 75.0 | 77.0 | 74.0 | 68.8 ± 2.4 | 56.1 |
| | 16384 | 100.0 | 90.5 | 61.0 | 60.0 | 81.5 | 26.9 | 28.3 | 28.5 | 72.0 | 58.0 | 70.0 | 75.0 | 62.6 ± 2.6 | |
| | 32768 | 100.0 | 90.7 | 62.0 | 64.0 | 78.5 | 20.9 | 21.3 | 27.1 | 36.0 | 27.0 | 60.0 | 74.0 | 55.1 ± 2.7 | |
| | 65536 | 100.0 | 72.4 | 61.0 | 51.0 | 76.5 | 21.7 | 23.1 | 32.0 | 19.0 | 14.0 | 55.0 | 71.0 | 49.7 ± 2.6 | |
| | 110000 | 99.0 | 58.1 | 52.0 | 44.0 | 70.0 | 19.3 | 13.9 | 26.4 | 6.0 | 9.0 | 64.0 | 67.0 | 44.1 ± 2.6 | |
| Qwen3 8B Thinking On | 8192 | 98.0 | 80.9 | 71.0 | 72.0 | 71.8 | 52.3 | 41.7 | 40.8 | 91.0 | 88.0 | 79.0 | 80.8 | 72.3 ± 2.3 | 52.1 |
| | 16384 | 95.0 | 62.0 | 56.0 | 58.0 | 67.8 | 34.4 | 29.9 | 24.3 | 78.0 | 64.0 | 75.0 | 72.0 | 59.7 ± 2.6 | |
| | 32768 | 90.0 | 46.3 | 51.0 | 61.0 | 62.3 | 20.2 | 19.7 | 22.1 | 58.0 | 56.0 | 76.0 | 70.0 | 52.7 ± 2.7 | |
| | 65536 | 71.0 | 30.1 | 41.0 | 42.0 | 47.5 | 16.4 | 13.9 | 9.9 | 36.0 | 37.0 | 68.0 | 62.0 | 39.6 ± 2.6 | |
| | 110000 | 70.0 | 27.3 | 41.0 | 34.0 | 43.5 | 15.7 | 11.7 | 12.2 | 35.0 | 14.0 | 64.0 | 65.2 | 36.1 ± 2.5 | |
| Qwen3 8B Thinking Off | 8192 | 98.0 | 81.9 | 63.0 | 61.0 | 68.8 | 64.3 | 30.0 | 39.9 | 79.0 | 59.0 | 71.0 | 74.0 | 65.8 ± 2.5 | 47.2 |
| | 16384 | 90.0 | 62.6 | 49.0 | 57.0 | 64.8 | 43.1 | 27.1 | 21.7 | 61.0 | 39.0 | 68.0 | 72.0 | 54.6 ± 2.6 | |
| | 32768 | 91.0 | 53.9 | 44.0 | 48.0 | 59.5 | 26.0 | 25.1 | 25.1 | 32.0 | 32.0 | 55.0 | 66.0 | 46.5 ± 2.6 | |
| | 65536 | 70.0 | 36.2 | 32.0 | 34.0 | 44.2 | 17.2 | 17.9 | 18.6 | 12.0 | 20.0 | 54.0 | 68.0 | 35.3 ± 2.5 | |
| | 110000 | 73.0 | 30.7 | 43.0 | 28.0 | 38.5 | 17.7 | 18.9 | 15.0 | 14.0 | 10.0 | 56.0 | 58.0 | 33.6 ± 2.5 | |
| Llama 3.1 8B | 8192 | 100.0 | 98.5 | 52.0 | 23.0 | 67.0 | 66.4 | 33.6 | 31.7 | 83.0 | 76.0 | 64.0 | 67.3 | 63.5 ± 2.5 | 51.3 |
| | 16384 | 100.0 | 93.5 | 29.0 | 12.0 | 59.2 | 54.9 | 29.6 | 38.2 | 71.0 | 68.0 | 56.0 | 63.3 | 56.2 ± 2.6 | |
| | 32768 | 100.0 | 87.1 | 22.0 | 8.0 | 50.5 | 44.0 | 25.3 | 30.5 | 85.0 | 69.0 | 51.0 | 60.0 | 52.7 ± 2.6 | |
| | 65536 | 97.0 | 75.6 | 23.0 | 6.0 | 38.0 | 30.0 | 23.0 | 31.1 | 75.0 | 67.0 | 38.0 | 57.3 | 46.8 ± 2.6 | |
| | 110000 | 91.0 | 62.6 | 20.0 | 0.0 | 26.5 | 22.4 | 18.1 | 15.1 | 42.0 | 52.0 | 44.0 | 53.5 | 37.3 ± 2.5 | |

