# OpenReview forum: "RULERv2: From Basic Retrieval to Complex Reasoning, A Bottom-Up Benchmark for Long-Context Evaluation"
_ICLR.cc/2026/Conference — Submitted to ICLR 2026_

### Official Review · Reviewer_9KeH · 2025-10-29

**Soundness:** 3
**Presentation:** 3
**Contribution:** 3
**Rating:** 6
**Confidence:** 4

**Summary:**

This paper (RULERv2) aims to address a core issue in the current evaluation of long-context language models: existing benchmarks typically combine multiple skills such as retrieval, aggregation, and reasoning into a single test. This makes it difficult for researchers to accurately identify the root weakness when a model fails (e.g., whether the failure stems from poor retrieval or flawed reasoning).

To solve this problem, the authors propose RULERv2, which spans three task domains (multi-key NIAH, multi-value NIAH, and multi-document QA). It gradually increases task complexity, transitioning to complex tasks that require retrieval, comprehension, counting, and multi-step reasoning. Its core innovation lies in providing a "structured, bottom-up diagnostic framework." RULERv2 is more than just a "performance ranking list"; it functions like a "diagnostic toolbox." It helps researchers accurately pinpoint which fundamental step a model is failing at—whether it is basic retrieval, multi-value retrieval, counting, or task decomposition.

**Strengths:**

The most prominent strength of this paper lies in its innovative systematic "bottom-up" diagnostic framework. It is not merely a performance benchmark, but a powerful "failure attribution" tool that can accurately diagnose whether a model fails in basic retrieval tasks (e.g., Easy tasks) or advanced strategy tasks (e.g., Hard tasks). By comparing the scores of explicit decomposition (Medium tasks) and implicit solving (Hard tasks), the paper uses data to clearly demonstrate for the first time that the core flaw of models lies in the lack of "autonomous task decomposition" capability, rather than the absence of the capability itself. It further reveals that Chain-of-Thought (CoT) can compensate for this flaw by helping models independently plan and decompose task steps. Finally, the evaluation design cleverly avoids the impact of data leakage and even converts it into evidence, proving that retrieval failure is a more fundamental bottleneck than knowledge deficiency. In summary, this is an evaluation tool with an elegant design, solid experiments, and profound insights.

**Weaknesses:**

While RULERv2 excels at diagnosing retrieval capabilities, its main drawback lies in the significant disconnect between its evaluation tasks and real-world needs. It overrelies on synthetic "needle-in-a-haystack" style tasks, which cannot effectively assess models' more critical real-world abilities of "information aggregation" and "comprehensive distillation". Secondly, since the evaluation data (e.g., MMLU) is highly likely to have been "memorized" by models, its "reasoning" component becomes hollowed out—reducing the evaluation to "retrieval plus recall" rather than genuine "retrieval plus reasoning". Finally, this also leads to an overly narrow definition of "fundamental skills": it one-sidedly equates long-context capabilities with "fact localization", while ignoring other equally important fundamental abilities such as "aggregation".

**Questions:**

1. Why isn’t "Aggregation" regarded as a foundational skill on par with Retrieval?
- Many long-context tasks in the real world—such as "summarizing a 100-page financial report" or "understanding the logic of a codebase"—rely not on "locating" a discrete "needle" as their foundational skill, but on "aggregating" a large amount of "diffuse" information distributed throughout the text.
- The paper criticizes other benchmarks (e.g., Longbench) for "mixing multiple skills" (because they include summarization tasks). However, RULERv2 itself fails to properly cite or discuss existing evaluation works that attempt to isolate the abilities of "aggregation" or "summarization."
- It seems to simply categorize "aggregation" under "advanced reasoning" (as mentioned in the introduction). Yet, I believe "aggregation" can fully be treated as another foundational skill—on par with "retrieval"—that requires "bottom-up" testing.
2. Evasion of the Limitation of "Hollowed-Out Reasoning"
- Using datasets like MMLU—whose content is highly likely to have been "memorized" by models—renders the "reasoning" step trivial.
The paper skillfully leverages this fact to prove that "retrieval failure is a more fundamental bottleneck."
However, it does not adequately address the new limitation this creates: the Hard tasks can no longer test "retrieval + genuine reasoning," but only "retrieval + autonomous decomposition."
- As a result, the paper also fails to properly cite relevant literature—specifically works that discuss how to construct "non-memorable" evaluations that truly require "on-the-fly reasoning."

In conclusion, this paper delves deeply into the vertical domain of "retrieval" and provides comprehensive citations. Yet, when arguing that "retrieval" is the sole or most important cornerstone, it does not sufficiently discuss relevant research on other foundational skills such as "aggregation."

---

> ### Author Response · Authors · 2025-11-23
> **Response (Part 1)**
>
> We thank the reviewer for the insightful and constructive feedback.
>
> > "Aggregation" as a Foundational Skill.
>
> We fully agree with the reviewer that aggregation is a critical and fundamental skill for long-context understanding. However, we view aggregation as an ability building upon retrieval based on this survey [1] (Sec 6.1.1). The definition of aggregation is to process pieces of information across multiple locations, recognize the connections between these pieces, and synthesize into coherent higher-level representations. This definition reveals an implicit dependency: a model cannot recognize meaningful connections or synthesize information if it cannot first reliably find and retrieve those discrete important pieces from the context.
>
> This hierarchy is further supported by the history of summarization evaluation. Before testing complex abstractive summarization, earlier benchmarks focused on extractive summarization as a foundational sanity check. By focusing on extractive summarization first, researchers can verify that models successfully retrieve and aggregate salient information throughout a document before attributing failures in abstractive summarization to synthesis deficits rather than information access problems.
>
> We position aggregation as hierarchically dependent on retrieval rather than as a parallel skill. RULERv2 explicitly tests aggregation through our multi-value NIAH task, which systematically evaluates aggregation capabilities across four difficulty levels.  This task can be viewed as a statistical aggregation. At the basic and easy levels, models must exhaustively retrieve all instances sharing the same key (e.g., all occurrences of "Question 123") and aggregate them into a unified response. At the medium and hard levels, models must retrieve scattered information, aggregate it into an ordered list, and perform ordinal selection (e.g., "the 2nd Question 123"), testing structured aggregation with positional reasoning. Our results reveal that aggregation remains a significant bottleneck, as even top performing models like GPT-4.1 achieve only 58% accuracy on the hardest aggregation task (Figure 5, bottom left). Critically, our four-level design enables precise failure diagnosis. GPT-4.1's drop from 98% (Easy: retrieval + basic aggregation) to 58% (Hard: ordered aggregation) demonstrates that the failure stems from structured synthesis rather than information access. This diagnostic precision is precisely what our bottom-up approach uniquely provides.
>
> > Criticism of existing benchmarks
>
> We appreciate the opportunity to clarify our critique of existing benchmarks. Our concern is not that benchmarks include aggregation, summarization, or other complex tasks, but that most test multiple skills simultaneously without isolation, making precise failure attribution impossible. When a model fails at a complex task, researchers cannot determine whether the failure stems from retrieval, aggregation, reasoning, or some combination of those.. We cite and appreciate prior works focusing on specific domains like ancestral tracing (reasoning) [2], counting stars (aggregation) [3], toy deduction (reasoning) [4], etc. However, these benchmarks typically evaluate complex instantiations without systematic difficulty progression. Our contribution is the systematic, bottom-up diagnostic structure. By progressing through four calibrated levels, we enable precise failure localization. Where prior works ask "Can models perform task X?", we ask "Which fundamental skill deficit prevents models from performing task X?" This diagnostic capability represents a novel methodological contribution that complements existing benchmarks.

---

> ### Author Response · Authors · 2025-11-23
> **Response (Part 2)**
>
> > "Hollowed-Out Reasoning" Limitation.
>
> We want to clarify that using memorized datasets like MMLU is not a limitation we are evading, but rather a deliberate methodological decision with clear scientific rationale. Our goal was to create a controlled experiment that isolates long-context-specific failures with surgical precision. When models achieve high accuracy on the base tasks in isolation, but exhibit drops when the same tasks are embedded in 128k contexts, we can definitively attribute failures to long-context handling rather than reasoning capability. Figure 5 (top right), demonstrates this approach generalizes across multiple base tasks (MMLU, GSM8K, MATH500, MBPP). We intentionally selected tasks with high baseline scores to ensure performance drops reflect long-context failures rather than task difficulty. GSM8K exhibits larger gaps precisely because its Easy-level retrieval score is lower, demonstrating how retrieval failures propagate to Medium and Hard tasks. On the other hand, if we had selected a very difficult task as the base task, we would simply observe low scores everywhere, making it hard to differentiate fundamental skill failures.
>
> We fully acknowledge the tradeoff of this design choice: it “hollows out” the test of genuine reasoning. However, this calculated exchange prioritizes diagnostic precision. It enables us to isolate unexpected failure modes (e.g., autonomous decomposition) that would remain hidden in benchmarks where retrieval and reasoning are confounded (as we cited in the related works). Moreover, our framework is task-agnostic. As models saturate on MMLU, we can substitute harder tasks while preserving the diagnostic paradigm.
> We thank the reviewer for this precise and valuable feedback. We have revised our paper and included the two major concerns in Limitations (section 8).
>
> Reference:
>
> [1] A Comprehensive Survey on Long Context Language Modeling\n
>
> [2] NeedleBench: Evaluating LLM Retrieval and Reasoning Across Varying Information Densities\n
>
> [3] Counting-Stars (★): A Multi-evidence, Position-aware, and Scalable Benchmark for Evaluating Long-Context Large Language Models
>
> [4] BABILong: Testing the Limits of LLMs with Long Context Reasoning-in-a-Haystack

---

### Official Review · Reviewer_6Pmo · 2025-10-31

**Soundness:** 3
**Presentation:** 4
**Contribution:** 3
**Rating:** 8
**Confidence:** 3

**Summary:**

The paper proposes RULERv2, a benchmark that evaluates the retrieval and reasoning capabilities of LLMs in long context settings. RULERv2 incorporates systematic tests of varying difficulty, from basic synthetic retrieval to multi-step reasoning, across three domains: multi-key NIAH, multi-value NIAH, and multi-document QA. These tasks extend the widely used Needle-in-a-Haystack tests by introducing multi-key and multi-value variants that isolate retrieval from reasoning, effectively decomposing different failure modes. Through experiments on 33 popular LLMs (including 7 closed-source and 26 open-weight models), the authors showcase consistent performance degradation with longer contexts and highlight weaknesses in retrieval and copying.

**Strengths:**

- The paper is well written, and the proposed benchmark is clearly presented and described in sufficient detail.
- The proposed task suite yields a fine-grained benchmark, where models are tested across tasks of varying difficulty and nature, from synthetic retrieval to multi-step reasoning.
- The empirical evaluation is comprehensive, spanning a diverse set of model families and scales, and provides detailed results for many widely used and state-of-the-art LLMs.
- The results in Sections 4 and 5 are particularly interesting. In particular, the observation that a retrieve-then-solve approach outperforms direct reasoning is insightful, as it implies that, in long-context settings, many models may underperform primarily due to inaccurate retrieval. Moreover, the paper provides strong evidence that the long-context capabilities claimed by several models are often overstated in practice. Overall, the results are insightful, and interesting.

**Weaknesses:**

- As the authors also point out in the limitations section, while the benchmark is comprehensive, it is mostly artificial. Therefore, although the results are interesting and comprehensive, they may not immediately translate to real world deployment scenarios.

**Questions:**

- In Figure 4, only the analysis of `Qwen3 235B Instruct 2507` is shown. Are the results qualitatively consistent across model families and scales?
- In the scaling width experiments of Section 5, how exactly do you define a "maximum score"?

---

> ### Author Response · Authors · 2025-11-23
> **Response**
>
> We thank the reviewer for the encouraging and thoughtful comments.
>
> > Although the results are interesting and comprehensive, they may not immediately translate to real world deployment scenarios.
>
> We fully agree that RULERv2 is primarily composed of artificial, synthetic tasks and that these results may not directly translate to end-to-end performance on all real-world scenarios. This was a deliberate design choice to achieve our benchmark's primary goal: to create a diagnostic test that can precisely isolate why models fail, rather than just finding if they fail on a complex and naturalistic task. In many real-world scenarios, a failure is ambiguous and it could stem from poor retrieval, faulty aggregation, or flawed reasoning. RULERv2's systematic bottom-up progression is specifically engineered to untangle these fundamental bottlenecks. We do not position RULERv2 as a replacement for naturalistic benchmarks, which are essential for assessing real-world deployment readiness. Instead, we see RULERv2 as a necessary complement, analogous to how unit tests complement integration tests in software development. It provides a rigorous, controlled way to measure fundamental skills (like retrieval or autonomous task decomposition) that are prerequisites for success on more complex, real-world applications.
>
> > Figure 4 analysis.
>
> Besides Qwen3 235B Instruct 2507, we provide additional results in Appendix C.1 with Qwen3 30B Instruct 2507, Qwen3 80B Instruct Next, and Seed OSS 36B Instruct. Most of the results are consistent with Figure 4. When increasing needle key length, we can see models show minimal degradation when the needle key length is short, but the scores degrade as the value length increases. Most critically, all models consistently degrade as the number of needles increases, demonstrating that existing LLMs struggle to reliably retrieve all relevant scattered information, regardless of their architecture, family, or scale.
>
> > Maximum score.
>
> In general, people will report pass@K as we increase the number of generations. However, pass@K only works for binary pass/fail metrics whereas RULERv2 uses a continuous score (as described in section 3). Therefore, we report max@K which is the maximum score across all the K generations, representing the model’s best possible performance The key finding is that max@K steadily increases (especially for larger models), indicating these models can occasionally produce correct answers but lack consistency. Meanwhile, maj@K shows minimal improvement, revealing that models tend to make similar errors repeatedly rather than generating diverse solution attempts. This suggests that better sampling strategies (e.g., reward models to select from diverse candidates) could improve performance more than simple majority voting.

---

### Official Review · Reviewer_MZgD · 2025-11-01

**Soundness:** 3
**Presentation:** 3
**Contribution:** 2
**Rating:** 4
**Confidence:** 2

**Summary:**

This paper introduces a benchmark that progressively increases task difficulty from basic synthetic retrieval to complex multi-step reasoning across three domains. It is used to evaluate 33 long-context models, and results uncover the limitations in current long-context capabilities that challenge existing claims of solved long-context understanding.

**Strengths:**

+ a new benchmark that progressively increases task difficulty from basic synthetic retrieval to complex multi-step reasoning across three domains
+ comprehensive evaluation using 33 long-context models to uncover the limitations in current long-context capabilities

**Weaknesses:**

- while the benchmark is good, the technical contributions are limited
- the findings reported in the evaluation are less insightful

**Questions:**

What are the challenges/difficulties in constructing the benchmark? What can be learned from the evaluation results to improve the performance of the models?

---

> ### Author Response · Authors · 2025-11-23
> **Response (Part 1)**
>
> We thank the reviewer for the thoughtful feedback. We agree that the primary goal of a benchmark paper is not just to present a new task, but to demonstrate that (1) its construction was a non-trivial technical challenge and (2) its application yields novel and actionable insights for the community.
>
> > Technical contributions are limited. What are the challenges/difficulties in constructing the benchmark?
>
> We respectfully argue that our primary technical contribution lies in the design of the systematic diagnostic framework itself. The core challenge in long-context evaluation is moving beyond aggregate scores to identify specific failure modes. RULERv2 addresses this through several non-trivial design decisions:
> - **Controlled Isolation of Failure Modes.** The core innovation of RULERv2 was our four-level progression (Basic -> Easy -> Medium -> Hard) where each transition isolates a single dimension of capability. For example, the only significant difference between our "Medium" (retrieve-then-solve) and "Hard" (single-step solve) tasks is removing the explicitness guidance for decomposition. This systematic design is a methodological contribution that allows us, for the first time, to cleanly separate a model's underlying capability from its ability to autonomously decompose complex tasks. Isolating failure modes is essential for diagnostic evaluation. When models fail on complex long-context benchmarks, practitioners need to know whether the root cause is inadequate retrieval, insufficient reasoning capability, or failure to autonomously decompose the task.Our systematic framework enables this precise attribution.
> - **Scalable and Flexible Design.** All our tasks scale from 8K to 1M+ tokens and support flexible base task substitution (e.g., swapping MMLU for GSM8K, Math500, or MBPP). This flexibility allows researchers to test long-context capabilities in their specific applications, while maintaining diagnostic precision. In summary, RULERv2's technical contribution is not merely task construction, but a reusable diagnostic methodology that enables precise failure attribution in long-context systems.

---

> ### Author Response · Authors · 2025-11-23
> **Response (Part 2)**
>
> > The findings are less insightful. What can be learned from the evaluation results to improve the performance of the models?
>
> We respectfully disagree that our findings lack insight. Our evaluation reveals actionable findings that contradict prevailing assumptions and provide concrete paths for model improvement. We highlight four key discoveries:
> - **The autonomous decomposition failure.** In Figure 5, we show that model scores consistently drop from medium (retrieve-then-solve) to hard (single-step solve) tasks, even when the base task accuracy remains high. This reveals that  failures are not due to insufficient reasoning capability, but rather the inability to autonomously decompose complex tasks. However, we find that Chain-of-Thought enables models to discover effective decomposition strategies, with the Hard+CoT performance approach that of Medium (Figure 6, top right). This finding suggests that some hard long-context tasks may not be easily solved by implicit single-step reasoning but need a clear step-by-step decomposition to achieve high scores.
> - **Retrieval is still unsolved.** Most long-context models tend to claim long-context ability based on high scores on basic retrieval tasks (NIAH). However, we find that top-performing open-weight models (e.g., Qwen3 235B Instruct 2507) still struggle on retrieval tasks when models must retrieve long-value needles (Figure 4, middle) and multiple needles (Figure 4, right). This is crucial because retrieval failures propagate to downstream reasoning. For example, a model cannot correctly answer a multi-hop question if it fails to retrieve all relevant facts. This suggests that model developers should prioritize robust retrieval, including multi-value and multi-target scenarios, as this fundamental capability can be a major bottleneck in complex tasks.
> - **Linear-attention hybrid models are still behind the full-attention models.** Open-weight linear-attention hybrid models consistently underperform full-attention models (Figure 3, middle). While hybrid architectures offer reduced compute and memory, this comes at a substantial accuracy cost for long-context tasks. Practitioners should benchmark architectural choices on realistic long-context evaluations before deployment. Our findings align with recent industry observations that prompted MiniMax M2's return to full attention [1], validating the importance of this tradeoff.
> - **Degradation is still happening when increasing lengths.** All tested models, even those claiming million-token context windows, exhibit systematic performance degradation with increasing context length (Figure 3). Even leading models like Gemini 2.5 flash experience a performance drop of 15% absolute at 1M context.. This is still an unsolved challenge in long-context modeling, and people tend to increase the task difficulty to showcase this issue.
>
> [1] Why Did MiniMax M2 End Up as a Full Attention Model?

---

### Official Review · Reviewer_jbNo · 2025-11-05

**Soundness:** 3
**Presentation:** 3
**Contribution:** 3
**Rating:** 4
**Confidence:** 5

**Summary:**

The paper presents RULERv2, a long-context benchmark with 12 tasks across three domains and four difficulty levels, aiming to evaluate LLMs from basic retrieval to complex reasoning. It tests 33 models (closed- and open-source) over contexts up to 1M tokens, analyzing performance degradation, model scaling, and test-time compute methods such as few-shot, chain-of-thought, and majority voting.

**Strengths:**

1.	**Comprehensive task coverage**.
The paper proposes a benchmark with 12 distinct tasks across three domains and four difficulty levels, which provides a relatively rich and systematic test suite. The breadth of evaluation makes the work look comprehensive and empirically grounded.
2.	**Diverse model evaluation.**
The experiments cover multiple model families — dense transformers, hybrid architectures, and Mixture-of-Experts (MoE) — including both closed- and open-weight models. This diversity offers a balanced empirical comparison.
3.	**Exploration of test-time compute scaling.**
The paper includes additional analyses such as majority voting, few-shot scaling, and reasoning step scaling. These experiments are useful for understanding whether increased inference-time compute improves performance (finding: only marginal gains).

**Weaknesses:**

**1. Limited conceptual novelty.**
While the paper frames RULERv2 as a “bottom-up” benchmark progressing from basic retrieval to complex reasoning, this design direction is *not entirely new*.
Earlier work such as **NeedleBench** has already proposed a clear decoupling between retrieval and reasoning, with additional control over information density and retrieval difficulty.
RULERv2’s main contributions appear to be **engineering-oriented extensions** — expanding task coverage, model variety, and systematic structure — rather than introducing a fundamentally new methodological idea.

**2. Potential data contamination.**
The benchmark relies on **HotPotQA** and **MMLU**, both of which are known to appear extensively in modern LLM pretraining corpora (e.g., Wikipedia-based content and academic QA).
Without explicit measures to control for **memorization effects**, it is difficult to determine whether the reported results genuinely assess **long-context reasoning**, or instead reflect **retrieval of memorized facts**.
Prior studies (e.g., *NeedleBench, Realistic vs. Synthetic Multi-Needle Tasks*) have shown notable performance drops once contaminated datasets are replaced by synthetic or unseen data.
A discussion or ablation to address this issue would strengthen the paper’s claims.

**3. Highly synthetic and procedural task design.**
Many tasks are **programmatically generated** and exhibit structured templates that differ from **real-world long-context scenarios** such as document summarization or multi-document reasoning.
While such synthetic setups facilitate controlled evaluation, they may also make the tasks **susceptible to overfitting** through supervised fine-tuning, potentially limiting generalization.
More **naturalistic benchmarks** (e.g., *LongBench v2*, Bai et al., ACL 2025) could complement RULERv2 by providing a stronger test of real-world long-context understanding. )

[1] NeedleBench: Evaluating LLM Retrieval and Reasoning Across Varying Information Densities. Transactions on Machine Learning Research.

[2] LongBench v2: Towards Deeper Understanding and Reasoning on Realistic Long-context Multitasks (Bai et al., ACL 2025)

**Questions:**

**1. On dataset contamination.**
Could the authors elaborate on whether potential **data overlap** with pretraining corpora (e.g., from MMLU or HotPotQA) has been analyzed or mitigated?
It would be helpful to understand how much of the observed performance may stem from **memorization** versus genuine **long-context reasoning**, perhaps through simple baseline comparisons or contamination checks.

**2. On task realism.**
Have the authors considered incorporating **more realistic long-context tasks**, similar in spirit to recent benchmarks that focus on **naturalistic document understanding**?
Such inclusion could make the benchmark more reflective of real-world long-context applications.

**3. On positioning and contribution.**
It would be valuable for the authors to clarify what they view as the **main conceptual advance** of RULERv2 relative to **prior long-context benchmarks**.
Since the idea of separating retrieval from reasoning has been explored before, highlighting what is **distinct or improved** in RULERv2 would help readers better appreciate its contribution and positioning. |

---

> ### Author Response · Authors · 2025-11-23
> **Response (Part 1)**
>
> We thank the reviewer for the valuable feedback and insightful comments.
>
> > Limited conceptual novelty.
>
> While we agree that prior work, such as NeedleBench, has explored the important decoupling of retrieval and reasoning, we respectfully clarify that RULERv2's primary contribution lies in introducing a systematic, four-level diagnostic framework that precisely identifies why models fail as tasks scale from basic retrieval to complex autonomous reasoning.
>
> The reviewer correctly notes that RULERv2's contributions include expanding task coverage and structure. We argue that this systematic structure represents our core methodological innovation: a bottom-up progression that allows for granular failure analysis, a capability missing in existing retrieval-reasoning benchmarks which also test model limitations by adding short-context skills into the retrieval tasks like NIAH [1][2][3][4].
>
> Our framework can enable precise diagnosis of failures, as shown in Section 2 (Page 2).
> - Failure at the Basic level indicates a fundamental inability to retrieve information from long contexts. For example, Qwen 2.5 7B 1M Instruct fails to locate multiple needles (Figure 5, bottom left).
> - Failure at the Easy level indicates an inability to retrieve and reconstruct information from realistic contexts. For example, all models degrade when copy-pasting realistic content in (Figure 5, top left).
> - Failure at the Medium level indicates deficits in specific capabilities such as counting, knowledge understanding, or QA reasoning. For example, Qwen3 235B-22B Instruct 2507 demonstrates poor counting ability (Figure 5, bottom left).
> - Failure at the Hard level indicates an inability to autonomously decompose complex tasks. For example, GPT 4.1 250414 cannot implicitly decompose hard tasks into retrieval-then-solve steps.
>
> Unlike NeedleBench, RULERv2 can help isolate whether multi-needle reasoning failures stem from retrieval, added skills (e.g., ancestral tracing in NeedleBench), or autonomous task decomposition. This diagnostic capability represents RULERv2's central methodological advance over existing long-context reasoning benchmarks that scale up NIAH difficulty without providing such granular analysis.
>
> Reference:
>
> [1] NeedleBench: Evaluating LLM Retrieval and Reasoning Across Varying Information Densities.
>
> [2] Babilong: Testing the limits of llms with long context reasoning-in-a-haystack.
>
> [3] Sequential-NIAH: A Needle-In-A-Haystack Benchmark for Extracting Sequential Needles from Long Contexts.
>
> [4] OpenAI MRCR: Long context multiple needle in a haystack benchmark

---

> ### Author Response · Authors · 2025-11-23
> **Response (Part 2)**
>
> > Potential data contamination.
>
> The issue of potential data contamination from MMLU and HotPotQA is a valid concern. We believe this is mainly applicable to our multi-key NIAH easy task, which asks the model to copy a MMLU question where only one such instance exists. We concede that for top-performing models, a high score on this specific sub-task could indeed result from recalling a memorized question.
>
> However, RULERv2's diagnostic structure is specifically designed to isolate and prove retrieval failures independent of memorization. Our evidence for this lies in the other task domains where simple memorization is not a viable strategy:
> - Our basic difficulty levels use purely synthetic, randomly-generated key-value pairs and document indices that are guaranteed contamination-free. Memorization of MMLU and HotPotQA cannot solve these tasks.
> - Our multi-value NIAH tasks require retrieving all scattered instances sharing the same index (e.g., finding four occurrences of "Question 123" across 128k tokens). No amount of MMLU memorization enables this, and models must perform genuine multi-location retrieval in order to solve this task. The significant performance drop on this task compared to the multi-key NIAH demonstrates a genuine failure in long-context retrieval.
> - Our multi-doc QA tasks require finding document indices through content matching or semantic search. A model that simply relies on memorizing whole HotPotQA documents (Wikipedia) cannot solve this task.
>
> Perhaps most importantly, we include “Base Task Performance” in Figure 5  to isolate the potential memorization issue.. This score measures a model's best-case performance to solve the MMLU or HotPotQA task in a zero-distractor, short-context setting, which includes any benefits from memorization. The significant gaps between base task performance and the scores on medium or hard show a clear failure of retrieval and task decomposition, not knowledge. In addition, to prevent bias in a single task, we also include the results of GSM8K, MATH500, and MBPP in our multi-key NIAH setup.
>
> Finally, while we agree there may be a potential memorization issue especially in the multi-key NIAH easy task, we view this as a diagnostic feature, not a bug. If a model scores high on multi-key NIAH easy while multi-key NIAH basic task and multi-value easy task show performance drop, this might indicate that the model is leaning on memorization and lacking fundamental retrieval and copying capabilities We will add this discussion to our limitations section and emphasize that while there might be some concern for contamination in one of the tasks, the overall benchmark is explicitly designed to be memorization-proof through synthetic generation, multi-index retrieval, or index-based search that decouples content knowledge from retrieval mechanics.

---

> ### Author Response · Authors · 2025-11-23
> **Response (Part 3)**
>
> > Highly synthetic and procedural task design.
>
> We fully agree that our benchmark's synthetic nature is a limitation for assessing real-world, end-to-end application performance. This was a deliberate design choice to achieve our primary goal: creating a diagnostic benchmark to precisely isolate why models fail, rather than just if they fail. As the reviewer correctly notes, naturalistic benchmarks like LongBench v2 are essential for real-world long-context understanding. We do not position RULERv2 as a replacement for these crucial benchmarks, but rather as a necessary complement that can provide the diagnostic precision needed for targeted development.
>
> Synthetic design enables three critical capabilities. First is scalability, as we can evaluate 8K to 1M tokens with automatic evaluation at scales where annotated realistic tasks rarely exist. Scaling these up even further as model context lengths improve is straightforward with synthetic tasks compared to real benchmarks. Second is precise attribution, as when models fail at a task like multi-doc QA hard, our progression isolates whether the cause is retrieval (Easy failure), reasoning (Medium vs. base task), or decomposition (Medium vs. Hard gap). Third is controlled experiments, as we can systematically vary context length, needle density, and complexity while holding other factors constant. Naturalistic multi-skill tasks cannot provide such diagnostic precision.
>
> Compared to RULERv1’s purely synthetic needles, our work bridges the synthetic-realistic gap through a hybrid approach: combining synthetic control structure with realistic content (MMLU, HotPotQA, GSM8K, etc.). The framework is also domain agnostic: practitioners can easily substitute any base task (coding repositories, legal documents, medical records, etc.) in order to diagnose long-context capabilities in their specific applications, while maintaining diagnostic precision. Our framework provides the control structure, while the task reflects real-world content and language.
>
> We sincerely thank reviewer for the thoughtful feedback, which has helped us clarify RULERv2's contributions and design choices. In response, we will make the following revisions:
> - Cite NeedleBench and clearly explain the difference in related works.
> - Add discussion of potential data contamination in the Limitations (section 8).
> - Clarify RULERv2's positioning as a diagnostic complement to naturalistic benchmarks
>
> We believe these revisions, combined with our detailed responses above, address the reviewer's concerns and better communicate RULERv2's distinct methodological contributions to the long-context evaluation landscape.

---

> ### Comment · Reviewer_jbNo · 2025-11-27
>
> I thank the authors for their detailed and clarifying response.
>
> The rebuttal effectively addresses my concerns regarding data contamination, particularly through the logic of "Base Task Performance" and the specific index-based retrieval mechanisms that cannot be solved by memorization alone. Regarding the synthetic nature and novelty, I agree with the authors that this is a reasonable trade-off to achieve precise diagnosis and scalability. While I still believe the ultimate goal for the field is to evaluate on more realistic long-context tasks (which prevents me from rating this even higher), I recognize the distinct value RULERv2 offers as a robust diagnostic tool.In light of this, I am raising my score to 6.

---

### Author Response · Authors · 2025-11-29
**Summary of discussion with all reviewers.**

We thank the reviewers for the insightful comments and constructive feedback. We are encouraged that the majority of reviewers acknowledged the soundness, presentation, and contribution of RULERv2. Through our active discussion regarding the concerns about contributions, data contamination, correlation with realistic tasks, and definition of fundamental long-context skills, we have received a positive score increase (Reviewer jbNo raised to 6) and strong support (Reviewer 6Pmo, Score 8). Below, we summarize our responses and revise our submission with updates highlighted in blue color.

> RULERv2 contributions.

We clarified for Reviewers jbNo and MZgD that our primary contribution is the systematic, four-level diagnostic framework. This design enables the granular isolation of failure modes, determining whether performance drops stem from retrieval,  reasoning, or any other implicit errors. Moreover, our evaluation of 33 models revealed critical insights: (1) Autonomous decomposition is hard for all models but can be discovered by Chain-of-Thought. (2) Basic synthetic retrieval is still unsolved even for top-performing open-weight models. (3) Hybrid linear-attention models currently lag behind full-attention models in long-context accuracy. (4) Performance degradation with increasing context length persists across all tested models, including those claiming million-token windows.

> Potential data contamination and following hollowed-out reasoning.

We addressed the concerns from Reviewers jbNo and 9KeH regarding the use of MMLU and HotPotQA. First, most of our tasks cannot be solved by memorized knowledge of a base task. Second, we have included "Base Task Performance" (Figure 5) to explicitly measure the gap between long-context and zero-distractor baseline. Third, we clarified that the selection of high performance base tasks is a deliberate design choice to isolate long-context failures without confounding them with the difficulty of the reasoning task itself.

> Lack of correlation with realistic tasks.

We acknowledge the feedback from Reviewers jbNo, 6Pmo, and 9KeH regarding the synthetic nature of our tasks. We have revised the paper to clarify that RULERv2 is positioned as a diagnostic complement to naturalistic benchmarks, not a replacement. Unlike purely realistic tasks, RULERv2 provides controlled failure modes analysis and scalable (increase length) and flexible (replace base task) evaluation.


> Lack of a clear definition of fundamental long-context skills.

Addressing Reviewer 9KeH’s concern that aggregation should be treated as a distinct fundamental skill, we clarified that: (1) we view aggregation as a capability that builds upon retrieval since a model cannot synthesize diffuse information without first successfully retrieving the discrete components; and (2) our Multi-value NIAH domain explicitly evaluates this capability, requiring models to locate information across multiple positions, recognize ordinal relationships, and generate an aggregated response.

---

### Meta-Review · Area_Chair_y9Px · 2026-01-13

**Summary:**

Across reviewers, there was broad agreement that RULERv2 is a well-executed and carefully presented benchmark with a comprehensive empirical evaluation. The main points of concern centered on conceptual novelty, the synthetic nature of the tasks, potential data contamination, and the scope of “fundamental” long-context skills being evaluated.

Several reviewers (jbNo, MZgD) questioned whether the benchmark introduced a fundamentally new idea beyond prior retrieval–reasoning decoupling work, viewing the contribution as more engineering- and scale-oriented. Relatedly, multiple reviewers (jbNo, 6Pmo, 9KeH) raised concerns that the highly synthetic, needle-in-a-haystack-style task design may limit correlation with real-world long-context applications such as summarization or document understanding.

Data contamination and “hollowed-out reasoning” were also recurring themes (jbNo, 9KeH), particularly due to the use of MMLU and HotPotQA, which may appear in pretraining corpora. Finally, Reviewer 9KeH raised a more conceptual critique regarding whether aggregation should be treated as a foundational long-context skill parallel to retrieval, rather than as a downstream capability.

Despite these concerns, reviewers generally acknowledged that the benchmark provides strong diagnostic value, and several explicitly noted that they would not object to acceptance even when scoring near the threshold.

**Reviewer Concerns:**

Concerns largely addressed by the rebuttal:

- Data contamination and memorization effects:
The authors provided a convincing clarification that most benchmark components are contamination-proof due to synthetic generation, index-based retrieval, and multi-value/multi-doc designs. The introduction of Base Task Performance as a control was persuasive to reviewers, with jbNo explicitly stating that this resolved their concern and raising their score accordingly.

- Conceptual positioning and novelty:
The rebuttal clearly articulated that the primary contribution is the four-level bottom-up diagnostic framework, emphasizing failure attribution rather than raw performance ranking. Multiple reviewers (jbNo, 9KeH) acknowledged this clarification as helpful and valid, even if some still viewed the novelty as incremental.

- Interpretation of findings:
For MZgD’s concern that the results were “less insightful,” the rebuttal successfully highlighted concrete, actionable insights (e.g., autonomous decomposition failure, retrieval as a bottleneck, architectural tradeoffs), which align well with the benchmark’s stated goals.

Concerns partially or not fully resolved:

- Disconnect from real-world long-context tasks:
While the authors convincingly framed RULERv2 as a diagnostic complement rather than a replacement for naturalistic benchmarks, several reviewers (notably 6Pmo and 9KeH) still viewed the lack of direct real-world task coverage as an inherent limitation. This concern was acknowledged but not fully mitigated.

- Scope of fundamental skills (aggregation vs. retrieval):
Reviewer 9KeH’s argument that aggregation should be treated as a parallel foundational skill was thoughtfully addressed, but remains more a matter of philosophical framing than empirical resolution. The authors’ hierarchical view (aggregation building on retrieval) is reasonable, yet some disagreement may persist.

- Limited perceived technical novelty (MZgD):
Although the rebuttal clarified the methodological contributions, this reviewer did not explicitly update their score, suggesting residual skepticism about the depth of technical innovation.

**Reviewer Scores:**

Reviewer jbNo:
Explicitly raised their score from 4 → 6 after the rebuttal, citing satisfactory resolution of data contamination and improved clarity on the benchmark’s purpose.

Reviewer 6Pmo:
Already strongly positive (score 8). The discussion reinforced, rather than altered, their assessment; no significant score change expected.

Reviewer 9KeH:
Likely to remain around 6. While the rebuttal engaged deeply with their concerns and clarified the aggregation and memorization issues, some philosophical disagreements about evaluation scope likely persist.

Reviewer MZgD:
No explicit post-discussion update was provided. As concerns about limited technical contribution may remain, the reviewer is unlikely to update the score.

Reviewer EKHh:
Did not provide a substantive review and thus does not factor into score changes.

---

### Decision · Program_Chairs · 2026-01-26

Reject